# Genomic changes in Kaposi Sarcoma-associated Herpesvirus and their clinical correlates

Jan Clement Santiago[1], Scott V. Adams[2], Andrea Towlerton[2], Fred Okuku[3], Warren Phipps[2,4], James I. Mullins[1,4,5]*

1 Department of Microbiology, University of Washington, Seattle, Washington, United States of America, 2 Fred Hutchinson Cancer Research Center, Seattle, Washington, United States of America, 3 Uganda Cancer Institute, Kampala, Uganda, 4 Department of Medicine, University of Washington, Seattle, Washington, United States of America, 5 Department of Global Health, University of Washington, Seattle, Washington, United States of America

* jmullins@uw.edu

**Data Availability Statement:** All relevant data are within the manuscript and its Supporting Information files.

## Abstract

Kaposi sarcoma (KS), a common HIV-associated malignancy, presents a range of clinico-pathological features. Kaposi sarcoma-associated herpesvirus (KSHV) is its etiologic agent, but the contribution of viral genomic variation to KS development is poorly understood. To identify potentially influential viral polymorphisms, we characterized KSHV genetic variation in 67 tumors from 1–4 distinct sites from 29 adults with advanced KS in Kampala, Uganda. Whole KSHV genomes were sequenced from 20 tumors with the highest viral load, whereas only polymorphic genes were screened by PCR and sequenced from 47 other tumors. Nine individuals harbored ≥1 tumors with a median 6-fold over-coverage of a region centering on K5 and K6 genes. K8.1 gene was inactivated in 8 individuals, while 5 had mutations in the miR-K10 microRNA coding sequence. Recurring inter-host polymorphisms were detected in K4.2 and K11.2. The K5-K6 region rearrangement breakpoints and K8.1 mutations were all unique, indicating that they arise frequently de novo. Rearrangement breakpoints were associated with potential G-quadruplex and Z-DNA forming sequences. Exploratory evaluations of viral mutations with clinical and tumor traits were conducted by logistic regression without multiple test corrections. K5-K6 over-coverage and K8.1 inactivation were tentatively correlated (p<0.001 and p = 0.005, respectively) with nodular rather than macular tumors, and with individuals that had lesions in ≤4 anatomic areas (both p≤0.01). Additionally, a trend was noted for miR-K10 point mutations and lower survival rates (HR = 4.11, p = 0.053). Two instances were found of distinct tumors within an individual sharing the same viral mutation, suggesting metastases or transmission of the aberrant viruses within the host. To summarize, KSHV genomes in tumors frequently have over-representation of the K5-K6 region, as well as K8.1 and miR-K10 mutations, and each might be associated with clinical phenotypes. Studying their possible effects may be useful for understanding KS tumorigenesis and disease progression.

**Funding:** This work was supported by National Institutes of Health (https://www.nih.gov) grants U54 CA190146 (WTP), K23 CA 150931 (WTP) and the University of Washington Centers for AIDS Research Retroviruses and Molecular Data Sciences Core (P30 AI027757; JIM). The funders had no role in study design, data collection and analysis, decision to publish, or preparation of the manuscript.

**Competing interests:** The authors have declared that no competing interests exist.

## Author summary

How chronic KSHV infection leads to the development of KS is poorly understood, however, evaluation of KSHV mutations found *in vivo* may assist this understanding. Here, targeted and whole genome sequencing of KSHV in multiple tumors from individuals in Uganda revealed an increase in copy number of the region of the genome encompassing the K5 and K6 genes or inactivation of the K8.1 gene in at least a third of the individuals. In an exploratory analysis these mutations were found more in tumors of nodular than macular morphotype and in individuals with fewer lesions. These mutations were unique across individuals, indicating that they arise frequently *de novo*. Point mutations in the miR-K10 micro-RNA sequence were weakly associated with lower survival. Occasionally, distinct tumors within an individual shared the same viral mutation, suggesting metastases or transmission of the aberrant viruses within the host. Clustering of genome rearrangement breakpoints were associated with potential G-quadruplex and Z-DNA forming regions in the viral DNA. The viral genetic differences uncovered and their associated clinical phenotypes call for targeted surveys of these genes to potentially establish a prognostic or treatment-selective biomarker and more studies into their roles in tumorigenesis.

## Introduction

Kaposi Sarcoma (KS) is an incurable endothelial cell disease [1,2]. The presence of spindle-like cells infected with Kaposi's Sarcoma-associated Herpesvirus (KSHV) is a hallmark of KS tumors [2,3], which can manifest in multiple forms and anatomic locations. KS tumors typically present as distinct patch, plaque or nodular morphotypes [4]. While KSHV is necessary for tumor formation [3], only about 1 in 10,000 infected individuals eventually develop KS [5]. KS is most often associated with HIV co-infection and AIDS, but it can occur even with high CD4+ T cell counts [6–9], and the risk for KS in antiretroviral drug treatment (ART) controlled HIV infections remain 800-fold greater than in the general population [10]. As a result, KS has become one of the most common cancers in sub-Saharan Africa [1,11–13]. In addition to HIV coinfection, many other factors can contribute to the development of KS [14,15].

Strain variation and de novo mutations in other DNA tumor viruses have been associated with varying disease risk and course [16,17]. However, only a few KSHV sequence variations have been correlated with disease. Variation in K1, the most variable gene when comparing KSHV isolates between individuals [18–20], has an unclear relationship with KS risk [21–24]. Polymorphisms in the KSHV-encoded microRNA (miR) region can lead to differential *in vitro* and predicted microRNA activity [25–27] and have been linked to higher risk for KSHV-associated multicentric Castleman disease (MCD) and KSHV-associated inflammatory cytokine syndrome (KICS) [28].

Potentially consistent patterns of diversity across the ~165-kb KSHV genome are beginning to emerge. For example, a study of tumor-derived near full-length KSHV genome sequences from Zambia revealed frequent nonsense mutations in the K4.2 gene, within the highly conserved central region of the genome [29]. That certain viral mutations were associated with tumor formation was suggested by our previous study in a cohort of 8 individuals that found that inactivation of the K8.1 gene as well as over-representation of a region near the internal repeat region IR1 was found in some tumors but not in oral swab-derived virus from the same individuals [30]. These mutations occurred in the context of near-perfect conservation elsewhere in the KSHV genome at the nucleotide sequence level. The effect of KSHV sequence

variation and *de novo* mutations are of potential significance to the biology and clinical course of KS. KSHV encodes immunomodulatory, angiogenic and anti-apoptotic factors, and its gene expression and replication are tightly regulated [31]. Hence, coupling KSHV polymorphisms observed *in vivo* in a larger cohort of individuals with detailed clinical data, as described here, may ultimately reveal insights into the pathogenic processes of KS.

## Methods

### Ethics statement

All study participants provided written informed consent. This protocol was approved by the Fred Hutchinson Cancer Research Center Institutional Review Board, the Makerere University School of Medicine Research and Ethics Committee (SOMREC), and the Uganda National Council on Science and Technology (UNCST).

### Study cohort and specimen collection

KS tumor biopsies and oral swabs were obtained from a prospective cohort of KS patients presenting to the Uganda Cancer Institute in Kampala, Uganda from 2012 to present. Briefly, chemotherapy- and antiretroviral therapy-naïve, HIV+ KS patients and endemic (HIV-) KS patients were enrolled and followed for up to 1 year over a series of clinical visits that included complete physical exam and recent health history. At baseline and 3 of 10 follow-up visits, KS tumors were biopsied as described in our previous study [30] and CD4+ cells and HIV viral load were assayed by quantitative RT-PCR. Participants from the cohort were chosen for this analysis to represent a range of clinical presentations, including HIV status, lesion morphotype and survival outcomes.

The following clinical and tumor traits were evaluated: gender, age, HIV load, CD4+ cell counts, plasma KSHV load, ACTG staging (T, S and I) [32], lesion prevalence, presence of any head, neck or oral lesions, and presence of lesions in the extremities. Tumor characteristics evaluated were morphotype (nodular, fungating or macular), size (greater or less than 1 cm diameter), anatomic area (head/neck, hard palate, oral excluding hard palate, back, chest/abdomen, groin/genitals, upper limbs, and lower limbs), and sampling time from first visit. Participant survival was followed as well.

### DNA preparation, PCR and KSHV gene copy number quantification

Genomic DNA was extracted using the QIAGEN DNeasy Blood & Tissue Kit (50) (Catalog # 69504) following manual single column extraction protocol per guidelines. All specimens were quantified using a Thermo Fisher Qubit Flurometer. Methods for droplet digital PCR (ddPCR) for gene copy number quantification, PCR screening of KSHV genes and for confirmation of breakpoint junctions have been described previously [30]. **S1 Table** lists the PCR primer sequences used in this study.

### Library preparation and sequencing

To obtain approximately 500-bp DNA fragments for sequencing, 10–20 ng/μL of DNA in 100 μL chilled TLE buffer (10mM Tris, pH8.0, 0.1mM EDTA) was sheared using a Covaris S2 Sonicator set to duty cycle 5%, 200 cycles per burst for a total of 30 seconds. Sheared DNA was purified using 1X volume of Agencourt AMPure XP Beads (Beckman Coulter Cat. # A63880) and eluted in 50 μL water. Library preparation (including end repair, A-tailing and adapter-ligation) was performed using the KAPA HyperPrep Library Preparation Kit (Cat. # KR0961/KK8503). Subsequently, the DNA was again purified with 1X volume beads and eluted in

50 μL of water. Samples were then divided into aliquots of <240 ng, if needed, prior to the next PCR step.

DNA libraries were subjected to pre-enrichment amplification with KAPA HyperPrep adapter primers (Cat# KR1317). The pre-enrichment PCR conditions were: 95˚C 4 mins; 5 cycles of 98˚C 20 sec, 60˚C 45 sec, 72˚C 45 sec; 72˚C 3 mins, 4˚C hold. Products were then pooled and purified as above with 1.2X volume beads and elution in 100 μL water, quantified with Nanodrop, and sizes assessed using a Bio-analyzer (Agilent DNA 7500).

KSHV sequences were enriched with biotinylated RNA baits [30], with the following modifications: Sample indexes for multiplexed sequencing were from the KAPA Single-Indexed Adapter Kit (Cat. # KR1317), primers for library amplification were *mws13* and *mws15* (**S1 Table**), and each DNA-RNA bait hybridization mixture was split into 2 PCR reactions and then amplified for up to 16 cycles. Products were cleaned with 1X volume XP beads. Sequencing was done on Illumina HiSeqX to obtain 150 bp paired-end reads.

## De novo assembly and analysis of KSHV genomes

Draft KSHV genomes were assembled de novo using SPAdes v3.11.1 [33] as reported previously [30], but omitting the initial step of trimming read ends. The setting -k 21,35,55,71,81,127 was used in SPAdes. Scaffolds produced from SPAdes were aligned to the KSHV reference genome GK18 (Genbank AF148805) for organization and orientation, inside the alignment viewer of Geneious (R11.1.5 for Mac OS, https://www.geneious.com). GK18 sequences were not used for refinement of draft genomes. Instead, sample read libraries were mapped to their respective draft genomes. The resulting consensus genome sequence was finished with GapFiller v1.1 using the sample paired-end read library [34]. For multiple sequence alignments of genomes, MAFFT (FFT-NS-i x1000, scoring matrix 1PAM/k = 2) implemented in Geneious was used. Most coding sequences (CDS) and non-coding RNA annotations were transferred from the GK18 genome via Geneious. Annotations for K8 and the variable genes K1 and K15 were transferred from the most homologous sequences among KSHV genomes GK18, Japan1 (Genbank LC200589) and ZM095 (Genbank KT271456). T1.4 RNA annotation was based on [35]. All CDS were inspected for indels.

Twenty new whole KSHV genomes sequenced from this study were uploaded to Genbank (accession numbers in parenthesis): U003-G (MZ923808), U004-F (MZ923809), U048-D (MZ923810), U048-E (MZ923811), U060-D (MZ923812), U099-D (MZ923813), U101-D (MZ923814), U108-B (MZ923815), U191-B (MZ923816), U210-B (MZ923817), U211-D (MZ923818), U215-D (MZ923819), U216-D (MZ923820), U219-D (MZ923821), U156-B (MZ923822), U156-C (MZ923823), U156-D (MZ923824), U156-E (MZ923825), U021-C (MZ923826), U021-E (MZ923827).

## Structural variation and integration breakpoint detection

The KSHV reference genome GK18 was appended as an extra chromosome to the human genome reference GRCh38 p12. Paired-end reads from each sample library was mapped to the appended human genome reference. Split reads and discordant read pairs extracted into separate alignment files were generated for each sample using SpeedSeq [36]. All potential rearrangement breakpoint locations, along with the number of supporting split and discordant paired reads, were tabulated from the SpeedSeq output files using LUMPY [37] and imported into Geneious as VCF annotations of genome GK18. Supporting reads of breakpoint positions in which the corresponding breakpoint have been identified in LUMPY were manually inspected in the Geneious alignment viewer for confirmation. In cases where LUMPY did not

output a breakpoint location for a region that had an abrupt read coverage change, split reads were identified using the Geneious read-mapping tool.

Potential breakpoints detected in LUMPY were accepted for analysis using the following criteria. 1) Breakpoints had to have at least 6 supporting split reads and at least 1 supporting paired read. For our earlier sequencing dataset that utilized duplex unique molecular identifiers [30], the threshold was lowered to 3 supporting split reads, the lowest that had PCR validation. 2) Breakpoints had to be at least 1 kb apart, approximately twice library insert sizes. 3) Identical breakpoint locations, which were found only in tumors from the same person, were counted as one. 4) Breakpoints of rearrangements wholly inside KSHV repeat regions were filtered out. 5) The coordinates of remaining breakpoints, along with the number of supporting reads and other output information from LUMPY and Geneious, were exported as a table.tsv file for analysis of the DNA sequence context in R (**S2 Table**).

## Analysis of DNA secondary structures and motifs

The following analyses were implemented with custom scripts in R with the Biostrings package [38] (https://github.com/MullinsLab/HHV8-non-B-DNA). Five kb sequences centered at the observed breakpoints (**S2 Table**) were taken from the KSHV genomes sequenced as part of this study. A permuted control dataset was generated by shuffling the sequences of the same 5 kb fragments (referred to as "control permuted" in figures). A randomized control dataset was generated from 5 kb sequences centering on an equivalent number of random positions along the GK18 genome, excluding TR sequences ("control random"). For each extracted window, the probabilities of cruciform, denaturation and Z-DNA formation at every base position were computed using *perl* software package SIST (Stress-Induced Structural Transitions)[39], set to the algorithm that considers competition among the three structures. Potential triplex and G-quadruplex formation was analyzed separately using R packages *triplex* [40] and *pqsfinder* [41], respectively. Both algorithms output a score at each position, which was normalized to a maximum of 300 [41] and 50, respectively. Non-B DNA probabilities and relative scores of only the central 1 kb were then plotted in R package *ggplot* for each observed breakpoint (**Supplementary Graphs**). Means of probabilities and relative scores were taken from equivalent positions across all breakpoint windows to create a plot of averages.

Comparisons of non-B DNA in observed breakpoints and control, through summed values and shortest distances, were done as follows. For each non-B DNA, probabilities or relative scores at positions ±200 bp from the observed breakpoints were summed. In addition, for each non-B DNA, the shortest distance of the observed breakpoint to a position with predicted non-B DNA probability or score >0.20 was taken from either direction, including 0. The summed values and shortest distances were calculated for the set of random breakpoints, and the differences between means of observed and random sets were tested using a Welch's T-test. To estimate the probability of attaining by random chance the observed mean G-quadruplex summed scores or higher, and the observed mean distances to G-quadruplexes or shorter, G-quadruplex summed scores and shortest distances were calculated from 1,000 simulations produced in R of sampling 31 random positions along the GK18 genome.

## Statistical analyses of clinical trait associations

Participant characteristics were summarized by median, interquartile range (IQR), or range as appropriate, or categorized and summarized by proportion. Tumor characteristics and viral genetic polymorphisms observed were classified into binary categories. Analyses were conducted at the tumor level. Logistic regression was used to estimate odds ratios (OR) with 95% confidence intervals (95% CI). Standard Errors (SE) were calculated using a cluster-robust

estimator to correct for correlation between tumors from the same participant. Exact logistic regression was used when logistic regression failed due to categories with no participants.

For analysis of participant survival at 1 year from study enrollment (OS), individuals were classified by whether they have at least one tumor with the observed mutation at baseline, and analysis was conducted at the participant level. Hazard ratios (HRs) for mortality with 95% CIs were estimated using Cox regression with robust standard errors.

Results were considered significant with two-tailed P<0.05, with no multiple tests corrections applied.

# Results

## Participant characteristics

A total of 67 KS tumor biopsies from 29 participants were included (Table 1). Twelve participants had only one tumor examined in this study, while 17 had more than one tumor

**Table 1. Study participant characteristics.**

| Study participant characteristics | N or *median* | % or (IQR) [a] |
|---|---|---|
| Number of participants | **29** | 100% |
| Gender: Male / female | **23** | 79% |
| *Age in years, median (range)* | *32* | *(23–78)* |
| KS stage [b]: | | |
| Tumor extent (T1) | **25** | 86% |
| Immune Status (I1) (CD4 <200) [c] | **12** | 41% |
| Systemic symptoms (S1) | **20** | 69% |
| HIV positive individuals: | **25** | 86% |
| *Median CD4 T-cell cells/mm$^3$ [c]* | *215* | *(85, 331)* |
| *Median plasma HIV RNA (log10 copies/ml) [c]* | *5.3* | *(5.1, 5.6)* |
| Plasma KSHV detected [d] | 27 | 96% |
| *Median plasma KSHV (log10 copies/ml) [e]* | *4* | *(3.7; 4.3)* |
| *Number of sites w/lesions (of 8 sites examined)* | *5* | *(3; 7)* |
| Any legs/feet lesions | **28** | 96.6% |
| Any arms/hands lesions | **24** | 82.8% |
| Any chest/abdomen lesions | **21** | 72.4% |
| Any groin/genital lesions | **15** | 51.7% |
| Any back lesions | **15** | 51.7% |
| Any head/neck lesions | **18** | 62.1% |
| Any oral hard palate lesions | **14** | 48.3% |
| Any oral lesions outside palate | **8** | 27.6% |
| Any head/neck/oral lesions | **18** | 62.1% |
| Any oral lesions | **15** | 51.7% |
| Any nodular lesions | **21** | 72.4% |

[a] Percent of all study participants; Interquartile range (25th percentile; 75th percentile)

[b] KS staging following AIDS Clinical Trial Group guidelines [32]: T = 1: tumors not limited to skin, with extensive oral, gastrointestinal & visceral KS; I = 1: CD4+ T-cell count <200/μL, NA if HIV (-); S = 1: Systemic illness (fever, night sweats, >10% weight loss, diarrhea for >2 weeks, opportunistic infections)

[c] Among HIV positive participants only

[d] % excludes 1 participant with missing information

[e] Among participants with detectable plasma KSHV RNA (> 150 copies/ml)

examined, up to 8 tumors in one person. Seven individuals had at least 4 tumors examined. Twenty-three participants were male, the median age was 32 years, and 25 were HIV+. According to the ACTG staging system [32], 25 of the KS cases were T-stage = 1, and 20 had systemic symptoms (S-stage = 1, see legend to Table 1). HIV+ participants had a median CD4 count of 215 cells/μL and a median HIV viral load of 2.1 x $10^5$ copies/mL.

## KSHV genomes in KS tumors often have higher representation of a subgenomic region near IR1

Our previous study described 5 unique tumor-associated KSHV genome aberrations, with 3 corresponding a 2 to 30-fold greater read coverage of a sharply delineated region close to IR1, compared to the rest of the viral genome [30]. To better determine the frequency of discordant read representation in the IR1 region in tumors, a total of 67 tumors from 29 individuals, including the 12 from [30], were screened for relative copy number elevation. Four short segments of the viral genome, within the K2, ORF16, ORF50 and ORF73 genes, were quantified by ddPCR. K2 and ORF16 are located near IR1, while ORF50 and ORF73 are located near the midpoint and 3' end of the KSHV genome, respectively (Fig 1A).

Of the 65 tumors with positive PCR results, 13 had ORF50 and ORF73 copy numbers that were a median of 6-fold less than K2 and ORF16, or below the limit of detection, consistent with an elevation in the IR1 region (Table 2). In contrast, only 2 tumors had K2 and ORF16 that were 2-fold less than ORF50 and ORF73 levels or below the limit of detection. Two tumors [30] had no DNA left for ddPCR (U003-C and U030-C), but their previous whole genome sequencing results did not reveal IR1 region read over-coverage [30]. In the remaining 48 tumors, copy numbers of the 4 genes were within a 1.5-fold range. Notably, when copy number variation was detected, it was not the case for all tumors sampled from an individual.

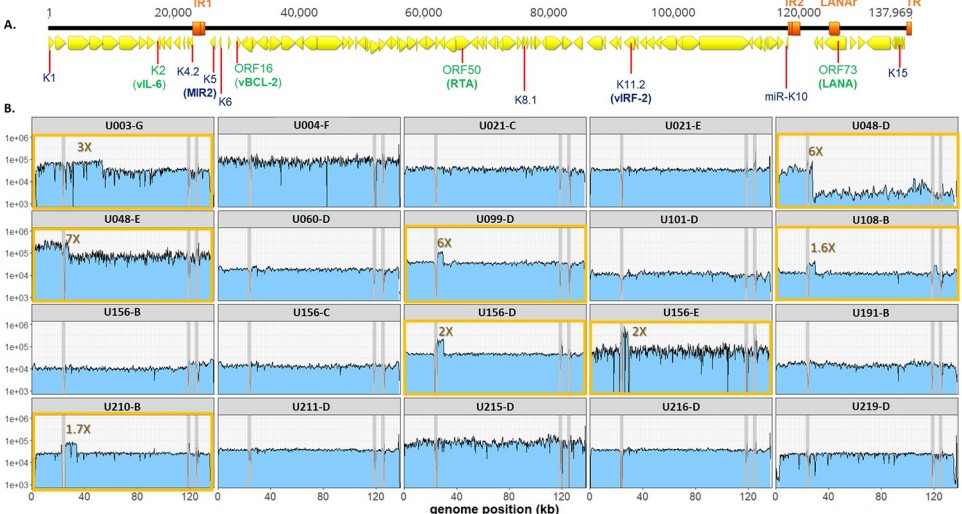

**Fig 1. Raw read coverage of KSHV genomes detected in 20 tumors from 16 individuals. (A.)** Linear schematic of KSHV genome. Yellow arrows represent open reading frames, as annotated in KSHV reference strain GK18. Orange bars represent the major repeat regions Internal Repeat 1 (IR1), Internal Repeat 2 (IR2), LANA repeat domain (LANAr) and Terminal Repeats (TR). Viral genes of note in this study are indicated, with their protein products in parenthesis. Those encompassing amplicons used for ddPCR quantitation are labeled in green. **(B.)** Raw read coverage of KSHV genome sequences from 20 tumor biopsies. The yellow boxes highlight the 8 tumors with the KSHV IR1 region overrepresented. The grey vertical lines represent the major repeat regions where sequence mapping was unreliable.

**Table 2. KSHV genome characteristics in 65 KS tumors.**

| Sample count | Sample ID [a] | Amplicon copy number (by ddPCR) [b] | | | | K2+ORF16 / ORF50+ORF73 [c] | KSHV genome (avg copies/μL) | Genome Sequenced | Genbank Accession No. | Overrepresented K5-K6 region [d] | Breakpoint K8.1 [e] | miR-K10 | K4.2 (bp) | K11.2 [f] 174-bp dup |
|---|---|---|---|---|---|---|---|---|---|---|---|---|---|---|
| | | K2 | ORF16 | ORF50 | ORF73 | | | | | | | | | |
| 1 | U003-B | 9,757 | 9,372 | 5,720 | 8,223 | 1.37 | 8,268 | | | | Breakpoint | G16A | 549 | no |
| 2 | U003-C | ND | ND | ND | 9,664 | ND | 9,664 | yes [30] | MT510648 | | Breakpoint | G16A | 549 | no |
| 3 | U003-E | 2,541 | 3,971 | 2,750 | 5,132 | 0.83 | 3,599 | | | | Breakpoint | G16A | 549 | no |
| 4 | U003-F | 3,702 | 2,063 | 2,618 | 3,119 | 1.00 | 2,876 | | | | Breakpoint | G16A | 549 | no |
| 5 | U003-G | 31,735 | 31,185 | 10,896 | 12,595 | **2.68** | 21,603 | yes | MZ923808 | yes | Breakpoint | G16A | 549 | no |
| 6 | U004-C | 1,243 | 1,274 | 1,205 | 998 | 1.14 | 1,180 | yes [30] | MT510663 | | Intact | WT | 549 | yes |
| 7 | U004-D | 4,433 | 4,466 | 4,543 | 4,290 | 1.01 | 4,433 | yes [30] | MT510665 | | TSS deletion | WT | 549 | yes |
| 8 | U004-F | 1,198 | 1,322 | 799 | 881 | 1.50 | 1,050 | yes | MZ923809 | | Intact | WT | 549 | yes |
| 9 | U007-B | 1,842 | 1,864 | 2,040 | 1,925 | 0.93 | 1,918 | yes [30] | MT510654 | | Intact | WT | 378 | yes |
| 10 | U008-B | 19,140 | 33,660 | 19,910 | 19,195 | 1.35 | 22,976 | yes [30] | MT510656 | yes | Intact | WT | 378 | no |
| 11 | U008-C | 16,995 | 17,325 | 19,170 | 15,235 | 1.00 | 17,181 | | | | Intact | WT | 378 | no |
| 12 | U008-D | 24,360 | 34,755 | 12,737 | 17,189 | **1.98** | 22,260 | yes [30] | MT510657 | yes | Intact | WT | 378 | no |
| 13 | U008-E | 7,744 | 7,871 | 6,617 | 5,451 | 1.29 | 6,921 | | | | Intact | WT | 378 | no |
| 14 | U008-F | 4,257 | 4,120 | 4,252 | 4,362 | 0.97 | 4,248 | | | | Intact | WT | 378 | no |
| 15 | U008-G | 3,839 | 4,164 | 3,889 | 4,257 | 0.98 | 4,037 | | | | Intact | WT | 378 | no |
| 16 | U008-H | 560 | 1 | 1,309 | 1,463 | *0.20* | 833 | | | | Intact | WT | 378 | no |
| 17 | U008-I | 8,701 | 9,059 | 7,541 | 8,080 | 1.14 | 8,345 | | | | Intact | WT | 378 | no |
| 18 | U020-B | 49,600 | 55,850 | 4,550 | 5,500 | **10.49** | 28,875 | yes [30] | MT510666 | yes | Intact | WT | 237 | no |
| 19 | U020-C | 3,658 | 5,033 | 145 | 139 | **30.62** | 2,243 | yes [30] | MT510667 | yes | Stop codon | WT | 237 | no |
| 20 | U020-E | 167 | 84 | 1 | 172 | 1.45 | 106 | | | | TSS deletion | WT | 237 | no |
| 21 | U020-F | 11,479 | 13,640 | 8,943 | 11,105 | 1.25 | 11,292 | | | | NPD | ND | NPD | NPD |
| 22 | U021-C | 5,691 | 5,943 | 6,867 | 6,101 | 0.90 | 6,151 | yes | MZ923826 | | Intact | C15T | 237 | no |
| 23 | U021-D | 1,190 | 1,134 | 1,348 | 1,680 | 0.77 | 1,338 | | | | Intact | C15T | 237 | no |
| 24 | U021-E | 15,792 | 16,558 | 23,083 | 20,050 | 0.75 | 18,871 | yes | MZ923827 | | Intact | C15T | 237 | no |
| 25 | U021-H | 1 | 424 | 454 | 826 | *0.33* | 426 | failed | | | Intact | C15T | 237 | no |
| 26 | U021-I | 234 | 311 | 312 | 205 | 1.05 | 266 | | | | Intact | C15T | 237 | no |
| 27 | U030-C | ND | ND | ND | 59,730 | ND | 59,730 | yes [30] | MT510670 | | Intact | WT | 369 | yes |
| 28 | U032-B | 476 | 520 | 494 | 466 | 1.04 | 489 | yes [30] | MT510652 | | Intact | WT | 549 | yes |
| 29 | U034-B | 1,920 | 1,887 | 1,870 | 1,793 | 1.04 | 1,868 | yes [30] | MT510659 | | Intact | WT | 378 | yes |
| 30 | U034-C | 13,083 | 13,335 | 8,148 | 6,552 | **1.80** | 10,280 | yes [30] | MT510660 | | Intact | WT | 378 | yes |
| 31 | U039-B | 5,313 | 5,502 | 4,946 | 4,799 | 1.11 | 5,140 | | | | Intact | WT | 378 | no |
| 32 | U048-B | 2,791 | 2,353 | 2,285 | 2,524 | 1.07 | 2,488 | | | | Intact | WT | 372 | no |
| 33 | U048-C | 549 | 84 | 79 | 71 | **4.22** | 196 | failed | | yes | Intact | WT | 372 | no |
| 34 | U048-D | 237,350 | 22,470 | 20,276 | 21,000 | **6.29** | 75,274 | yes | MZ923810 | yes | Intact | WT | 372 | no |
| 35 | U048-E | 7,142 | 717 | 602 | 592 | **6.58** | 2,263 | yes | MZ923811 | yes | Intact | WT | 372 | no |
| 36 | U060-C | 26,460 | 25,830 | 24,990 | 25,725 | 1.03 | 25,751 | | | | Intact | G16A | 369 | yes |
| 37 | U060-D | 24,885 | 25,305 | 26,040 | 24,360 | 1.00 | 25,148 | yes | MZ923812 | | Intact | G16A | 369 | yes |
| 38 | U062-B | 1,765 | 1,991 | 2,127 | 1,846 | 0.95 | 1,932 | | | | Intact | WT | 369 | no |
| 39 | U062-C | 3,266 | 3,287 | 2,510 | 2,407 | 1.33 | 2,868 | | | | Intact | WT | 369 | no |
| 40 | U066-C | 1,107 | 1,147 | 1,192 | 974 | 1.04 | 1,105 | | | | Intact | WT | 378 | yes |
| 41 | U094-B | 1,829 | 1,846 | 1,855 | 1,419 | 1.12 | 1,737 | | | | Intact | WT | 378 | no |

(*Continued*)

**Table 2.** (Continued)

| Sample count | Sample ID [a] | Amplicon copy number (by ddPCR) [b] | | | | $\frac{K2+ORF16}{ORF50+ORF73}$ [c] | KSHV genome (avg copies/μL) | Genome Sequenced | Genbank Accession No. | Overrepresented K5-K6 region [d] | Breakpoint K8.1 [e] | miR-K10 | K4.2 (bp) | K11.2 [f] 174-bp dup |
|---|---|---|---|---|---|---|---|---|---|---|---|---|---|---|
| | | K2 | ORF16 | ORF50 | ORF73 | | | | | | | | | |
| 42 | U099-D | 999,999 | 999,999 | 140,910 | 195,090 | **5.95** | 584,000 | yes | MZ923813 | yes | Stop codon | WT | 549 | yes |
| 43 | U101-D | 5,061 | 5,271 | 4,767 | 3,150 | 1.31 | 4,562 | yes | MZ923814 | | Intact | C15T | 549 | no |
| 44 | U106-B | 3,108 | 3,108 | 4,001 | 4,106 | 0.77 | 3,581 | | | | Intact | WT | 549 | no |
| 45 | U108-B | 53,655 | 50,610 | 45,360 | 45,150 | 1.15 | 48,694 | yes | MZ923815 | yes | Stop codon | WT | 549 | no |
| 46 | U108-H | 1 | 1,067 | 1 | 1 | **534.00** | 268 | failed | | yes | NPD | NPD | NPD | NPD |
| 47 | U146-C | 3,003 | 2,982 | 2,489 | 2,128 | 1.30 | 2,651 | | | | Intact | WT | 549 | no |
| 48 | U156-B | 3,045 | 3,119 | 2,877 | 4,862 | 0.80 | 3,476 | yes | MZ923822 | | Intact | WT | 378 | no |
| 49 | U156-C | 8,526 | 9,240 | 7,875 | 9,356 | 1.03 | 8,749 | yes | MZ923823 | | Intact | WT | 378 | no |
| 50 | U156-D | 163,083 | 159,750 | 164,500 | 171,583 | 0.96 | 164,729 | yes | MZ923824 | yes | TSS deletion | WT | 378 | no |
| 51 | U156-E | 1,010 | 1,001 | 275 | 789 | **1.89** | 769 | yes | MZ923825 | yes | TSS deletion | G16A | 378 | no |
| 52 | U156-G | 387 | 413 | 379 | 287 | 1.20 | 367 | | | | Intact | WT | 378 | no |
| 53 | U156-H | 3,262 | 3,471 | 2,206 | 2,822 | 1.34 | 2,940 | | | | Intact | WT | 378 | no |
| 54 | U191-B | 21,389 | 21,515 | 19,541 | 21,431 | 1.05 | 20,969 | yes | MZ923816 | | Intact | WT | 237 | no |
| 55 | U191-C | 9,261 | 9,534 | 9,597 | 9,807 | 0.97 | 9,550 | | | | Intact | WT | 237 | no |
| 56 | U191-D | 623 | 662 | 455 | 676 | 1.14 | 604 | | | | Intact | WT | 237 | no |
| 57 | U191-E | 324 | 599 | 372 | 411 | 1.18 | 427 | | | | Intact | WT | 237 | no |
| 58 | U191-F | 39 | 1,091 | 43 | 26 | **16.38** | 300 | failed | | yes | NPD | NPD | NPD | NPD |
| 59 | U210-B | 14,700 | 31,675 | 15,125 | 12,917 | **1.65** | 18,604 | yes | MZ923817 | yes | Stop codon | WT | 237 | no |
| 60 | U211-D | 87,583 | 91,667 | 89,750 | 84,167 | 1.03 | 88,292 | yes | MZ923818 | | Intact | WT | 378 | no |
| 61 | U215-D | 36,752 | 51,538 | 4,100 | 4,150 | **10.70** | 4,135 | yes | MZ923819 | yes | Intact | WT | 237 | yes |
| 62 | U216-D | 121,333 | 130,083 | 121,000 | 110,250 | 1.09 | 20,667 | yes | MZ923820 | | Stop codon | C15T | 237 | no |
| 63 | U217-D | 2,914 | 3,008 | 3,988 | 3,289 | 0.81 | 3,300 | | | | Intact | WT | 375 | no |
| 64 | U218-D | 13,075 | 13,467 | 12,500 | 12,125 | 1.08 | 2,792 | | | | Intact | WT | 378 | no |
| 65 | U219-D | 19,188 | 19,600 | 32,958 | 23,875 | 0.68 | 3,905 | yes | MZ923821 | | Intact | WT | 378 | no |

Key

ND = not determined; tumor extracts were exhausted in previous study [30]

WT = wildtype; database consensus

NPD = no product detected

[a] Adults with KS contributed multiple tumor biopsies from distinct lesions to this study and were anonymized with a "U" number. The following letter is the tumor identifier

[b] 1 = below the limit of detection; 999,999 = over the limit of detection

[c] Ratio of ddPCR counts of (K2 and ORF16) over (ORF50 and ORF73). Ratios higher that 1.5 are bolded, ratios lower than 0.5 are italicized

[d] Integrating both ddPCR screening data and whole genome sequencing read coverage data

[e] Breakpoint = breakpoint of a genomic rearrangement in KSHV, interrupting the K8.1 coding sequence [30]; TSS deletion = transcription start site deletion; Stop codon = contains a nonsense mutation that truncates the open reading frame

[f] Duplication of the central 174-bp domains of K11.2 (vIRF-2)

Twenty consensus KSHV genome sequences were obtained from 10 individuals with >2-fold higher copy numbers of K2 and/or ORF16 compared to ORF50 and/or ORF73, including 1–3 additional tumors from the same individuals and 1 tumor each from 6 individuals with no discordant ddPCR results, choosing tumors with the highest viral DNA load. Greater than 99.9% of the KSHV genome aside from the 3 repeat regions was successfully sequenced from 20 of 24 tumors, including those from as low as 769 average genome copies per μL as estimated by ddPCR (**Table 2**). Eight of the 20 tumors had elevated read coverage that encompassed a region between genome positions 26,000 and 32,000 (**Fig 1B**). In 5 of these (U099-D, U108-B, U156-D, U156-E, U210-B) the spike in read coverage was sharply confined to this region. Read over-coverage was also found in 2 tumors (U108-B and U156-D) in which copy numbers of the K2 and ORF16 regions were not elevated in the ddPCR screen (**Table 2**). This was due to the amplicons used in ddPCR being just outside the regions of over-coverage. Taking the ddPCR copy number variation and whole genome sequencing results together, a total 16 of 65 tumors, from 10 of 29 individuals, were determined to have IR1 region over-representation.

The precise boundaries of the overrepresented regions, as well as any genomic aberrations not apparent from read coverage changes, were determined by mapping supporting split reads (see Methods). Two tumors from participant U048 (U048-D and U048-E) were found to have the same breakpoint. Two of 4 tumors from U156 (U156-D and U156-E) also had identical breakpoints, while 2 of this participant's other tumors had relatively even read coverage across the genome (**Fig 1B**). No breakpoint was found joining KSHV and human sequences together, which would have indicated integration of KSHV genomes into human chromosomes.

Putative breakpoint junctions in the original tumor DNA from U048-D were validated by PCR across breakpoints followed by sequence confirmation. Breakpoint junctions were found that joined IR1 to TR sequences, K6 to TR sequences (**Fig 2**) and connecting IR1 to K6 sequences in the opposite orientation (**S1A–S1C Fig**). The breakpoint junction sequences suggest multiple rearrangements of the K5-K6 region with at least one inversion, but the structure was not further characterized due to the length of the repeats and high GC content in IR1.

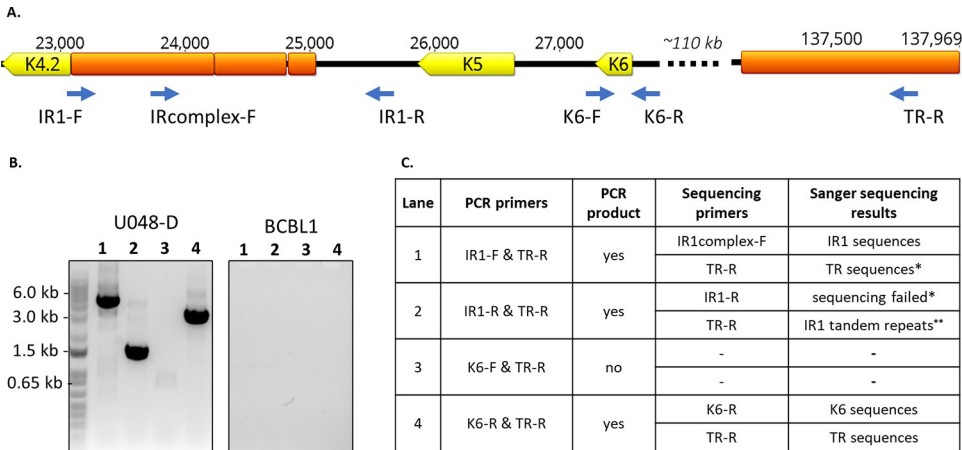

**Fig 2. Confirmation of breakpoints connecting IR1 and K6 to TR sequences in tumor U048-D. A.** The region of the KSHV genome from K4.2 through K6 and a distant TR unit sequence is shown. ORFs are shown in yellow, repetitive sequences in orange, and primers with blue arrows. **B.** PCR products were separated on a 0.8% agarose gel, and visible bands were extracted and sequenced using the Sanger method. DNA from the BCBL1 cell line was used as control, and no visible PCR products were produced. (PCR products from primer pairs IR1-F & IR1-R and K6-F & K6-R are shown in S1 Fig). **C.** Summary of results. * = Primarily due to ambiguous and low quality base calls. ** = Result truncated at 200bp.

Lastly, minor variants in U048-D containing an 82-bp inversion between ORF9 and ORF10 coding sequences were detected and confirmed (**S2 Fig**).

## A 2.2-kb region encompassing K5 and K6 corresponds to the minimal region of overrepresentation

In the 32 tumor-derived KSHV genomes sequenced between this and our previous study [30], nine unique KSHV genome aberrations had an abrupt, ≥1.5-fold read coverage over-representation in a subgenomic region near IR1. Strikingly, all encompassed a 2.2kb segment downstream of IR1 that included the K5 and K6 genes (**Fig 3**). The recombinant KSHV strain BAC36 had been reported to have read overrepresentation in the same region [42] (**Fig 3**). The T1.4 long non-coding RNA was found in 6, and the PAN long non-coding RNA was included in 8 of 9 cases.

The average length of the genome segments with read coverage overrepresentation was 17.6 kb (**Fig 3**), or ~13% of the 137 kb KSHV genome excluding the terminal repeats (TR). Given ten random 17.6-kb windows of the KSHV genome sequence, that all 10 would include the 2.2-kb K5-K6 region is highly improbable ($2.3 \times 10^{-38}$). No other non-repeat region >3 kb in size had >1.5-fold read over-coverage in any of the 32 tumors in our cohort with whole KSHV genomes sequences determined. Through either ddPCR screening or whole viral genome sequencing, the K5-K6 region overrepresentation was found in at least one tumor from 10 individuals (**Table 2**), approximately one-third of the cohort of 29 individuals. Of note, participant U020 had two tumors with different breakpoints [30] yet both contained the minimal overrepresented region (**Fig 3**).

There were 4 clusters of breakpoints defining the endpoints of each overrepresented region: Near K4.1-K4.2, the IR1 region, minor internal repeats between K7 to ORF16, and ORF18-ORF19 (**Fig 3**). The 5' breakpoints from two individuals (U156-B and U210-B) were 0.2 kb apart within genes K4.1 and K4.2. Three 5' breakpoints were within the second GC-rich tandem repeat family of IR1, with one more ~500 bp downstream. Three 3' breakpoints were within 0.3 kb at or near the minor internal repeats. Four 3' breakpoints were within OR18 or ORF19, two of which, from 2 different individuals, were only 3 bp apart (U008-B and U210-B).

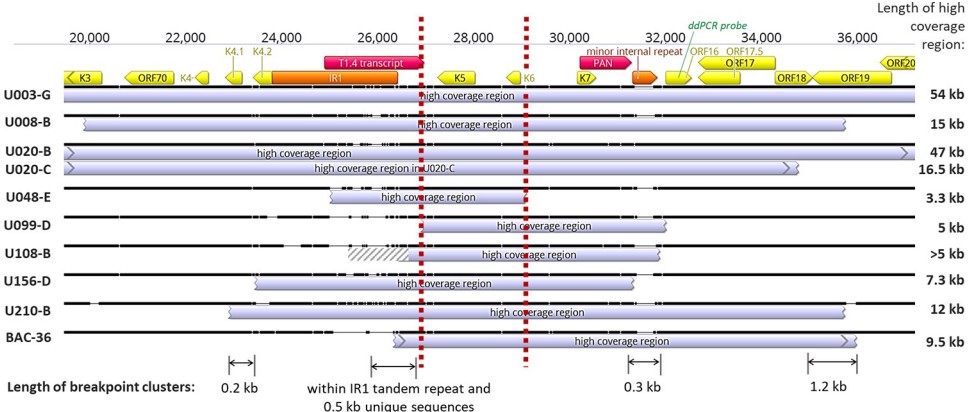

**Fig 3. Overlap in KSHV IR1 overrepresentation regions.** Sequence alignment of regions with excess read coverage near IR1 (showing those with identical breakpoints only once). Location and orientations of ORFs are in yellow, long non-coding RNAs in red, and repetitive sequences in orange. Gene region used for ddPCR probe for screening is in green. The 'minor internal repeat' is composed of the 13-bp sequence TGGGATGGGGGTG repeated 4 to 13 times. Vertical dashed red lines demarcate the minimal high coverage region. Indicated below are the sizes of the 4 breakpoint clusters identified. Breakpoint coordinates of the high coverage regions shown are indicated in S2 Table.

## KSHV rearrangement breakpoints are associated with potential G-quadruplexes

Chromosomal rearrangement breakpoints are prone to cluster at fragile regions, typically sequences that form non-B DNA structures [43]. Genomes of gamma-herpesviruses are enriched, more than expected from nucleotide composition, with GC-rich DNA motifs known to induce double-stranded breaks that undergo repair and recombination in B-cells [44,45]. We therefore assessed whether KSHV rearrangement breakpoints were associated with these DNA features.

To identify associations between KSHV rearrangement breakpoints and potential non-B DNA sequences, sequences ±500 bp from all 31 breakpoints identified here and in other studies [30,42,46] (S2 Table) were scanned for sequence motifs associated with the following 5 non-B DNA structures: cruciforms, Z-form DNA, AT-rich local melt regions, mirror repeats or triplexes, and G-quadruplexes (G4) (S3 Fig). This window length was chosen because breakpoints have previously been associated with G4 DNA up to 500 bp distant [47]. Control windows were generated by permuting the sequences in the observed breakpoint windows.

Breakpoints were most common at or near predicted Z-DNA and G4 sequences (Fig 4A and 4B). The 2 breakpoints 3 bp apart in ORF19 were within a high probability Z-DNA sequence. On average, Z-DNA probabilities had a local peak at the observed breakpoint although a nearby peak was also generated from the random control dataset (Fig 4C). G4 scores also peaked at the breakpoint position but not in the control data (Fig 4D). Cruciform DNA was not predicted to occur in any of the breakpoint or control windows.

To quantify their association with breakpoints, non-B DNA probabilities or relative scores within 200 bp in both directions from the 31 observed breakpoints were summed, and the mean summation in this set was compared to that of 31 random breakpoints (S2 Table) using a Welch 2-sample T-test (Table 3). Among the 5 non-B DNA motifs analyzed, G4 had the largest difference between the means of the observed and random datasets (p = 0.004). As another measure, distances in either direction from each breakpoint to the nearest position with >0.20 non-B DNA probability or relative score were counted, and the mean distances were compared with that of control breakpoints (Table 3). The closest distances to local melt regions and regions with relative G4 scores >0.20 were significantly smaller for the observed breakpoints than for the random dataset (p = 0.024, p = 0.0007, respectively).

To get a more robust assessment of association with predicted G4 structures, 1000 simulations of 31 randomly sampled points along the GK18 genome were made to generate a normal distribution of means. The probability of attaining equal or higher means than the observed mean G4 summed scores was $1.4 \times 10^{-6}$, and the probability of attaining equal or lower than the observed mean distances to G4 was 0.0009.

## Potentially inactivating mutations are common in the K8.1 gene in tumors

We previously observed inactivating mutations (truncation, inversion, transcription start site deletion) in the K8.1 gene in KS tumors [30]. In the current study, 5 of 20 whole KSHV genomes sequenced had inactivating mutations in K8.1 (tumors U108-B, U156-D, U156-E, U210-B and U216-D). In contrast, none of the 85 other KSHV coding segments had intra-host differences in more than one person, even though three-quarters of the KSHV coding sequences are larger than K8.1. Additionally, not all individuals with a K8.1 gene mutation had the same mutation in all of their tumors. To better assess the frequency of K8.1 mutations, a 1.4 kb region encompassing the K8.1 coding sequence and promoter region was determined by PCR and sequencing in all 65 tumors from 29 individuals. Parallel analysis of a 250 bp K12 gene sequence served as sample control.

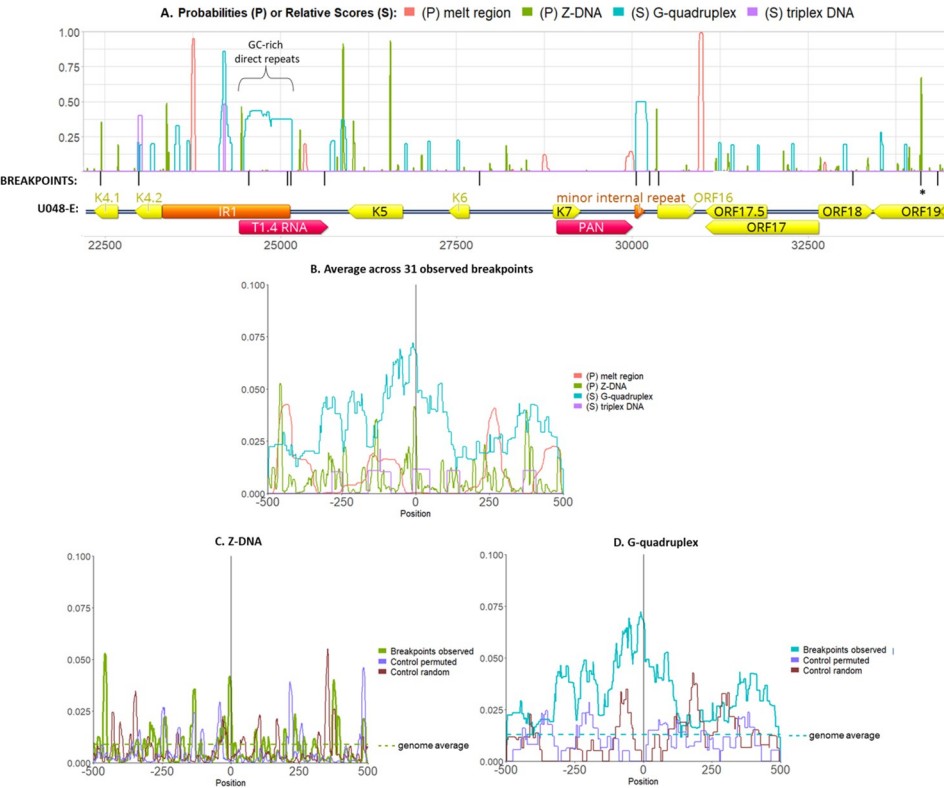

**Fig 4. Association of breakpoints to non-B-DNA.** The probabilities and relative scores of 4 non-B-DNA structures (local melt region, Z-DNA, G-quadruplex and triplex DNA) were graphed along the genome of KSHV isolate U048-E. Shown is a genome segment including the overrepresented region including K5 and K6. The black ticks mark the breakpoints observed, and below are the annotations for ORFs (yellow), repeat regions (orange) and long non-coding RNAs (red). The asterisk at ORF19 denotes 2 breakpoints 3 bp apart found in 2 different individuals. G-quadruplex scores were normalized to 300, near the human genome maximum (https://pqsfinder.fi.muni.cz/genomes), and triplex scores were normalized to 50, the maximum found. **(B.)** Probabilities or relative scores were plotted in DNA sequences +/- 500 bp of 31 observed breakpoints (not including those at TR) and averaged across all windows. Score normalizations are as in panel A. Of note, all averages were below 0.10 since probabilities and scores were 0 within most breakpoint windows. **(C.)** Z-DNA probabilities and **(D.)** G-quadruplex scores in control permuted sequences and 31 control random windows. The genome-wide averages are shown with the horizontal broken lines.

A total of 9 unique K8.1 mutations were detected in 8 individuals (**Table 2**, **Fig 5A**), including five tumors with nonsense mutations in the second exon, and three with 28–32 bp deletions between the K8.1 core promoter and the K8.1 coding sequence, including the transcription start site (**Fig 5A**). These deletions were all unique, and one included the first base of the K8.1 coding sequence (U020-E in **Fig 5B**). Finally, participant U003 had a genomic inversion interrupting the second K8.1 exon and extending to TR sequences [30]. U020 had 2 tumors with different K8.1 mutations–a nonsense mutation in U020-C and a transcription start site deletion in U020-E (**Fig 5A**).

## Polymorphisms in miR-K10, K4.2 and K11.2

Promoter region deletions, indels and nonsense mutations in the 85 KSHV genes other than K8.1 were compiled from all 20 tumor-derived KSHV whole genomes sequenced here and the 12 from our previous study [30]. No other promoter region deletions or nonsense mutations were found. Seven different indels were identified, 4 of which were in-frame (**Table 4**). Only

**Table 3. P-values of non-B-DNA summed probabilities or scores, and distances to breakpoints.**

Sum of probabilities or relative scores of non-B-DNA structures within +/-200 bp of breakpoint*

| Non-B-DNA | Mean obs. | Mean random | Diff of means | 95% CI lower | 95% CI upper | P-value |
|---|---|---|---|---|---|---|
| melt | 0.041 | 0.121 | -0.08 | -0.26 | 0.10 | 0.380 |
| Z-DNA | 1.577 | 1.839 | -0.26 | -2.16 | 1.63 | 0.782 |
| cruciform | 0.000 | 0.000 | 0.00 | 0.00 | 0.00 | NA |
| triplex | 0.933 | 0.300 | 0.63 | -0.82 | 2.08 | 0.385 |
| G4 | 16.094 | 2.640 | 13.45 | 4.60 | 22.31 | 0.004 |

Average distance to closest non-B-DNA structure with probability / relative score >0.2

| Non-B-DNA | Mean obs. | Mean random | Diff of means | 95% CI lower | 95% CI upper | P-value |
|---|---|---|---|---|---|---|
| melt | 450.258 | 498.200 | -47.94 | -89.12 | -6.765 | 0.024 |
| Z-DNA | 278.516 | 327.560 | -49.04 | -154.86 | 56.769 | 0.357 |
| cruciform | >500 | >500 | NA | NA | NA | NA |
| triplex | 433.871 | 442.200 | -8.33 | -77.06 | 60.405 | 0.809 |
| G4 | 308.645 | 460.480 | -151.83 | -235.30 | -68.371 | 0.0007 |

* Probabilities are shown for melt, Z-DNA and cruciform; relative scores are shown for triplex and G4.

one, an A insertion in an A homopolymer run in K11.2, could be identified as an intra-host mutation, as it was found in only 1 of 4 tumors from U156 (**Table 4**).

Intra-host differences between matching sample-consensus genomes were determined for 10 individuals who had whole KSHV genomes sequenced from more than one sample, including from the oral swabs sequenced previously [30]. Six nonsynonymous point mutations were

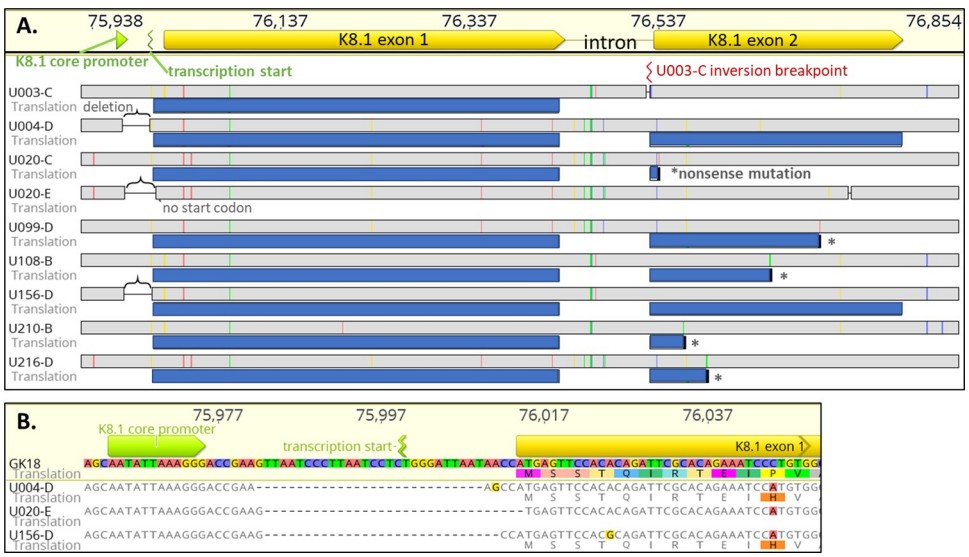

**Fig 5. Inactivating mutations observed in the K8.1 gene. (A.)** Highlighter plot of an alignment of K8.1 gene and promoter mutations found in this study, showing identical mutations from the same person once. Indicated above are locations of the K8.1 core promoter, transcription start site and open reading frame. Sequence coordinates are for reference isolate GK18. Gray bars in line with tumor IDs represent their nucleotide sequences, while blue bars below represent translations of open reading frames to their fullest extent. Colored ticks represent nucleotide differences from the GK18 sequence (red: A, blue: C, yellow: G, green: T), brackets indicate transcription start site deletions, asterisks indicate nonsense mutations, and red mark corresponds to the inversion breakpoint location in U003 tumors. **(B.)** Sequence alignment of the transcription start site deletions. Indicated below nucleotides are amino acid translations if there are open reading frames. Highlighted bases and amino acids correspond to those differing from the consensus.

**Table 4. Coding sequence mutations, excluding those in K8.1, K4.2 and K11.2 and rearrangement breakpoints.**

| Gene | GK18 annotation | Mutation | Tumor(s) | Intra-host differences |
|------|-----------------|----------|----------|------------------------|
| | | **In frame deletions** | | |
| ORF34 | Related to HHV-5 UL95 | 3 nt | U004-C,D,F | 3 of 3 tumors |
| ORF47 | Envelope glycoprotein gL; herpesvirus core gene UL1 family | 15 nt | U004-C,D,F | 3 of 3 tumors |
| ORF50 | Transactivator; RTA, induces switch from latent to lytic infection | 24 nt | U099-D | 1 of 1 tumor |
| K9 | Interferon regulatory factor, vIRF-1 | 18 nt | U216-D | 1 of 1 tumor |
| | | **Frameshifts** | | |
| ORF4 | Complement control protein; membrane protein; contains four SCR domains; KCP | 23 nt deletion, truncated ORF | U048-D,E | 2 of 2 tumors |
| K7 | IAP-like inhibitor of apoptosis; contains hydrophobic domain; vIAP | 1 nt homopolymer deletion, extended ORF | U216-D | 1 of 1 tumor |
| K11.2 | Interferon regulatory factor, vIRF-2 | 1 nt homopolymer insertion, truncated ORF | U156-D | 1 of 3 tumors |
| | | **Intra-host non-synonymous point mutations** | | |
| ORF11 | herpesvirus dUTPase | T396P | U020-C | 1 of 3 tumors and 1 oral swab |
| K3 | E3 ubiquitin ligase, membrane protein MIR1, downregulation of MHC1; ORF12 | F88L | U020-C | 1 of 3 tumors and 1 oral swab |
| ORF25 | major capsid protein; herpesvirus core gene UL19 family | Q594K | U020-B | 1 of 3 tumors and 1 oral swab |
| ORF32 | tegument protein; herpesvirus core gene UL17 family; DNA packaging | R56Q | U004-D | 1 of 3 tumors and 1 oral swab |
| ORF63 | tegument protein; herpesvirus core gene UL37 family | T848A | U032-B | 1 of 1 tumor and 1 oral swab |
| K15 | LAMP; signal transducing membrane protein | A290P | U004-D | 1 of 3 tumors and 1 oral swab |
| | | **Intra-host synonymous point mutations** | | |
| ORF8 | envelope glycoprotein gB; herpesvirus core gene UL27 family | nucleotide: C762AT | U021-E | 1 of 2 tumors |
| K12 | Kaposin A, hydrophobic membrane protein; contains microRNA K10 | nucleotide: G126A (K12); G16A (miR-K10) | U003-B,C,E,G; U156-D | 4 of 4 tumors and 0 of 3 oral swabs; 1 of 4 tumors |

found in 6 different genes (**Table 4**). Three instances of synonymous point mutations were detected, including 2 that were the same K12 G126A nucleotide substitution found in 2 different individuals. This mutation was found in 1 of 4 tumors from U156, and in all 4 tumors (**Table 4**) but none of the 3 oral swabs (**S3 Table**) from participant U003. This mutation impacted the overlapping microRNA K10 sequence and is designated miR-K10 G16A (position C118082T in the GK18 genome). To determine the extent of miR-K10 diversity in this cohort, PCR and sequencing was conducted on all 65 tumors (**Table 2**) and 18 oral swabs (**S3 Table**). Five of 29 individuals had either a miR-K10 C15T or G16A mutation, all in tumors, while the remaining sequences had the database consensus nucleotide in those positions.

Truncations of the K4.2 coding sequence were reported in 12 of 16 adults with KS in Zambia [29]. In our studies, truncations of K4.2 were found in 16 of 22 individuals, resulting from frameshifts and premature stop codons (**Fig 6A**). However, no intra-host differences were detected. A 174-bp duplication in the central domain of K11.2 was present in 8 of the 22 individuals we studied (**Fig 6B**) and in 12 of the 16 genomes from Zambia [29] (a partially overlapping set with those that had a truncated K4.2). To determine the overall extent of K4.2 and K11.2 diversity in our cohort, both genes were sequenced following gene-specific PCR in all 65 tumors (**Table 2**). K4.2 was truncated in 23 of the 29 individuals, while the central 174-bp duplication in K11.2 was found in 9 of 29. No intra-host differences were found.

Among all 96 KSHV genomes sequenced to date, 3 predominant length polymorphisms of K4.2 were found: 31 were full-length at 549 or 546 bp, 47 had a truncation to 378 or 369 bp, and 18 had a truncation to 237 bp or shorter (**S4 Table**). Of the 94 KSHV genomes with a

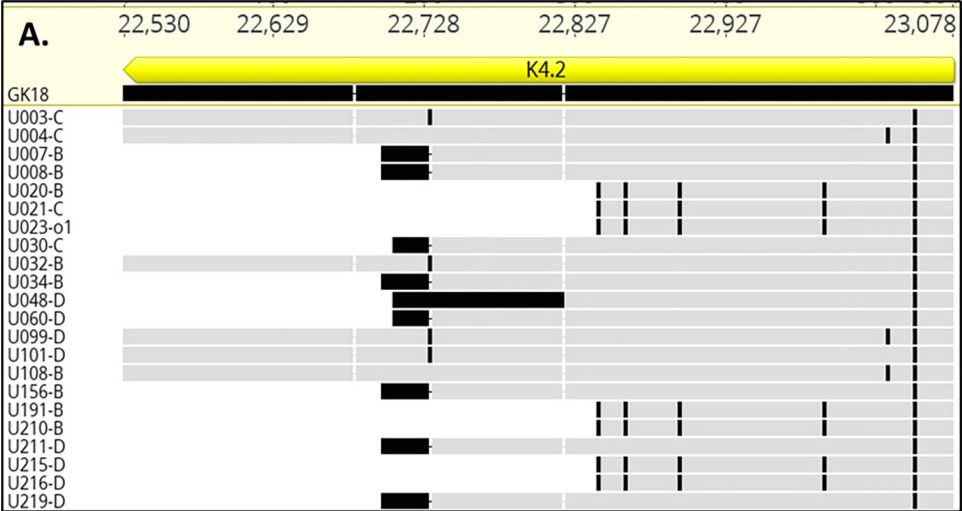

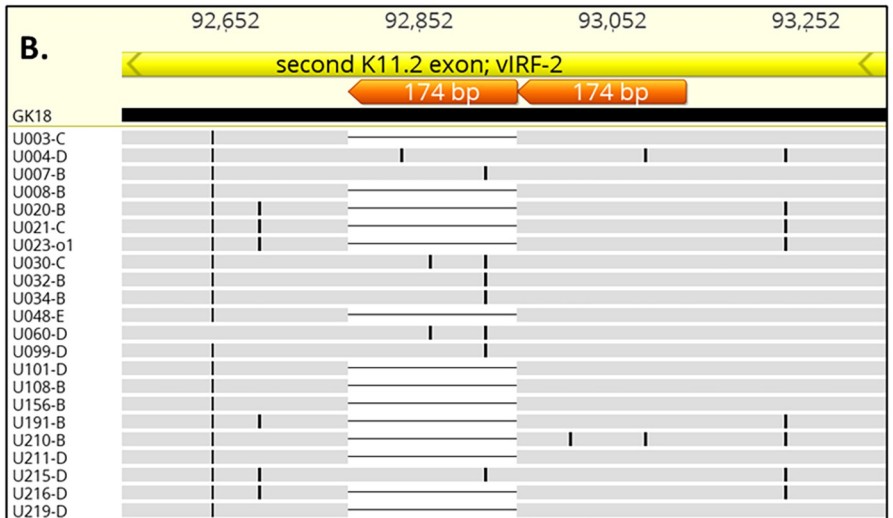

**Fig 6. Polymorphisms in K4.2 and K11.2.** Alignment of predicted K4.2 (**A.**) and K11.2 (**B.**) amino acid sequences in whole KSHV genomes sequenced in this study aligned against the GK18 reference, showing 3 length genotypes. Black marks indicate amino acid differences from GK18. The 22 sequences in (A) are representative of K4.2 length polymorphisms found in the 98 KSHV genomes sequenced to date. The 174-bp repeats found in K11.2 are in orange.

complete K11.2 gene, 76 had the 174-bp duplication (**S4 Table**), including 31 with 1 or 2 nucleotide differences in the duplicated sequence. A phylogenetic network was created from all sequenced KSHV genomes to date (**Fig 7**). Notably, the K4.2 and K11.2 polymorphisms did not coincide fully with major KSHV phylogenetic groupings [20] (**Fig 7**; **S4 Table**). Since data assessing K5-K6 over-coverage and K8.1 inactivation is not available from most published sequences, we are unable to assess linkage of these characteristics with KSHV phylogenetic groupings.

## KSHV mutations associated with disease course and tumor characteristics

An exploratory statistical analysis was conducted for associations between the observed mutations or genetic polymorphisms and clinical traits (**S5 Table**) or tumor characteristics (**S6 Table**). The mutations observed here were evaluated as genetic markers: whether K5-K6 region had read over-coverage over 1.5X, whether K8.1 gene was inactivated, whether

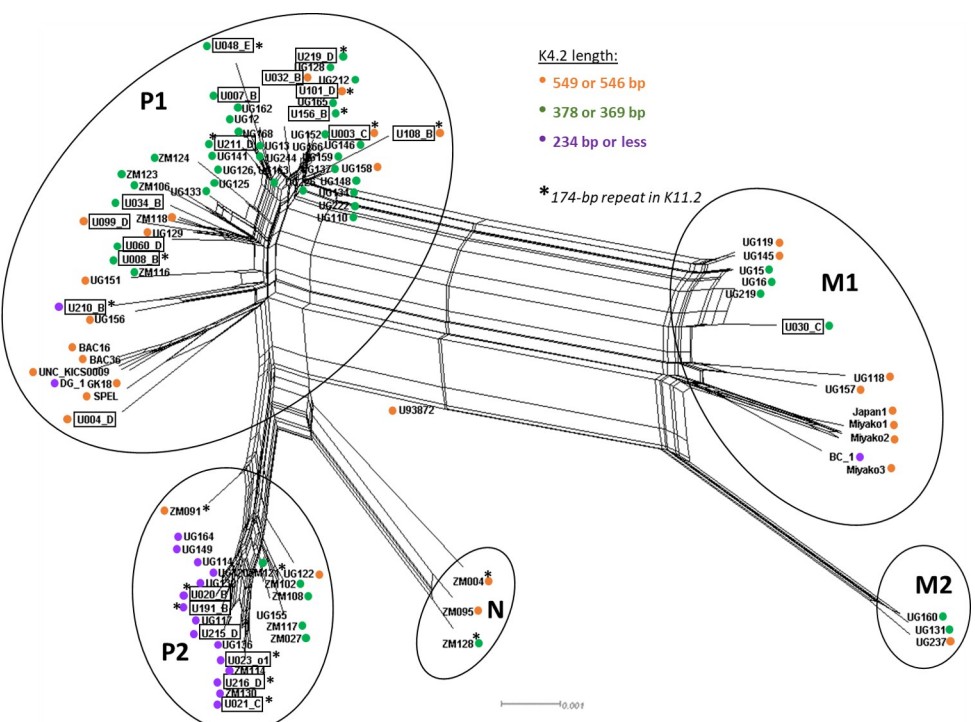

**Fig 7. Phylogenetic network of 96 whole KSHV genomes, showing K4.2 and K11.2 genotypes.** Figure generated using SplitsTree4, excluding gap sites. Major KSHV phylogenetic groupings [20] are indicated by the circles. Whole KSHV genomes sequenced in this study are boxed. Their K4.2 and K11.2 genotypes are marked as in the legend.

miR-K10 had point mutations relative to database consensus, whether K4.2 was truncated, whether K11.2 174 bp central domain was duplicated (**Fig 8, S7 Table**), whether any of these mutations significantly occurred together (**S8 Table**), and whether these mutations were associated with any clinical or individual tumor traits (**S7 and S9 Tables**).

When examining for associations between the described KSHV mutations, none was found except for between the K4.2 truncation and the K8.1 inactivation, which were inversely correlated (OR = 0.12, p = 0.008, **S8 Table**). In univariable analysis, K5-K6 over-representation and K8.1 inactivation were significantly more common in nodular compared to macular lesions (**Fig 8**; OR = 20.2, p<0.001 and OR = 6.38, p = 0.005, respectively; **S7 Table**). Both were also more common in participants that had fewer KS lesions (OR = 0.61, p = 0.01 and OR = 0.57, p = 0.024, respectively) and no head, neck and oral lesions (OR = 0.13, p = 0.01, and OR = 0.02, p<0.01, respectively), although individuals with no head, neck and oral lesions typically had fewer lesions overall. Since all K4.2 truncations observed removed the putative transmembrane region, K4.2 genotype was classified into having either a full-length or truncated coding sequence. A truncated K4.2 was less common in nodular compared to macular lesions (OR = 0.28, p = 0.028). The K11.2 174-bp domain duplication was potentially more common among female participants (OR = 7.56, p = 0.048). The miR-K10 mutations C15T and G16A were less prevalent in lesions >1 cm (OR = 0.20, p = 0.010) (**S7 Table**) and potentially associated with lower survival rates (HR = 4.11, p = 0.053, **S9 Table**).

## Discussion

This study is the most extensive screening and whole genome sequencing to date of KSHV in tumors. From a total of 65 KS tumors from 29 individuals, the K8.1 mutations, the over-

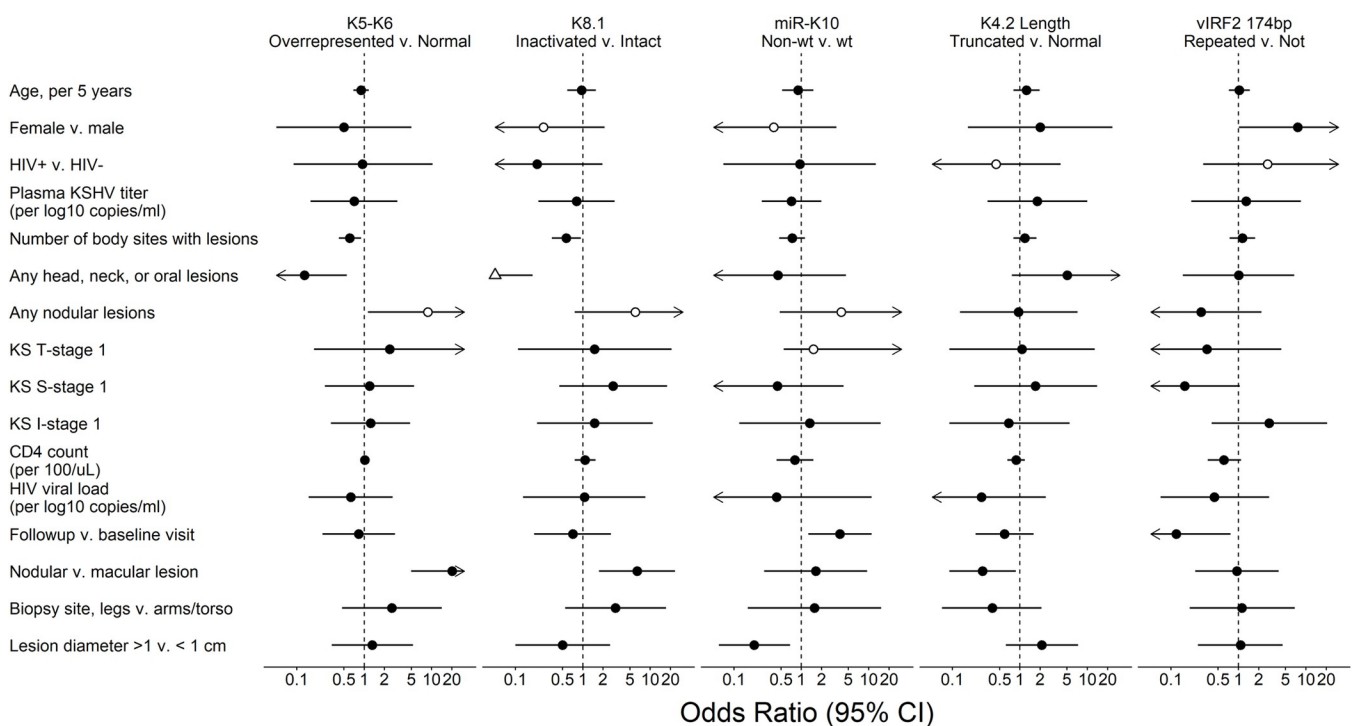

**Fig 8. Associations between observed KSHV mutations and clinical traits.** Forest Plot of the odds ratio (OR) of participant and tumor characteristics with KSHV mutations from regression analysis of KS tumors from 29 participants, univariable analysis. Filled symbols were estimated with logistic regression and hollow symbols with exact logistic regression. Triangle indicates the OR point estimate is outside of the plotted range. Arrows indicate that the plotted range extends beyond the scale bar below.

representation of the K5-K6 gene region, and to a lesser extent, the mutations in the miR-10 sequence, occurred at highly significant frequencies than would be expected by chance. Viral genomic structures within tumor biopsies were heterogeneous, although in multiple instances, genomes with aberrant structures were more abundant than apparently intact viral genomes. Additionally, these variations were associated with certain clinical manifestations of KS tumors.

## Recurring KSHV Genome Aberrations in KS Tumors

KSHV sequences derived from tumors and tumor cells often have large genomic aberrations. The first whole KSHV genome sequenced had a 33-kb region duplicated in the TR region [48], and KSHV genomes with major deletions have also been found by PCR screening in some KS tumors and cell lines [49]. In a cohort of 16 individuals with KS in Zambia, 4 harbored KSHV genomes with uneven read coverages indicative of genomic aberrations [29]. Our previous study characterized large inversions, deletions and duplications present in KS tumors from 4 of 8 individuals [30].

Rearranged KSHV genomes involving read coverage over-representation of regions 3.3 to 53 kb in length was present in 10 of 29 individuals, in all cases including the K5 and K6 genes. Read over-coverage ranged from 1.5 to 6,000-fold with a median of 6-fold. The real frequency of K5-K6 region overrepresentation could be higher, since only one or a minority of tumors from each participant were available for study, and the ddPCR screen quantified genomic segments outside the minimal high coverage region. Indeed, whole genome sequencing identified over-coverage of the K5-K6 region in 2 tumors that was not identified in the ddPCR screen. It would be useful to directly quantify via a targeted amplification approach such as ddPCR to

compare K5/K6 gene copy numbers to the rest of the KSHV genome when looking for this phenomenon in the future.

The high-coverage regions found in 9 whole KSHV genomes sequenced encompass at minimum a 2.2-kb sequence containing the K5 and K6 genes. However, they are not among the known KSHV oncogenes [50] and thus a direct role in tumorigenesis is not likely. K5 is a viral ubiquitin E3 ligase best known to ubiquitinate and degrade MHC1 from the cell surface [51]. It is a membrane protein expressed upon primary infection and reactivation [52,53] and at low levels during latency [54]. Another KSHV ubiquitin E3 ligase, K3, is far more efficient in degrading MHC1 [51], but K5 modulates more immunoreceptors than K3, targeting also adherins, ligands and co-stimulators for cytotoxic T and NK cells, cytokine receptors, restriction factors and members of the SNARE, ephrin, plexin and other receptor families [51,55,56]. One downregulated ephrin receptor of note is EphA2 [55,56], one of the receptors employed by KSHV for entry into endothelial cells [57]. Amplification of K5 might suggest that cells may become less visible to cytotoxic T and NK cells and refractory to KSHV superinfection as EphA2 expression on host cell surface is depleted by K5. Lytic gene K6 encodes vMIP-I, a viral homolog of cellular cytokine MIP1α (CCL3). vMIP-I is a selective, high affinity agonist for chemokine receptor CCR8 and has angiogenic and chemotactic effects *in vitro* [58–60]. The inclusion of K6 in the minimum region is puzzling, given that vMIP-I is not readily detectable in KS lesions by immunohistochemistry [61].

Other rearranged KSHV genomes may have been present but could not be confirmed since low numbers of split reads connecting non-adjacent regions of the KSHV genome were indistinguishable from PCR chimera artifacts. Additionally, while another gamma-herpesvirus, EBV, can be occasionally found to be integrated into its host's chromosomes [62], no evidence of integration of KSHV sequences to human ones was found.

## Possible role of defective viral genomes

It is striking that genome fragments containing K5 and K6 were amplified or retained in nearly all genome rearrangements that were detected, becoming the majority or only detected genome form in some tumors. Nearly all K5-K6 subgenomic fragments connect to IR1, a KSHV origin of lytic replication [63,64], and to TR sequences, which have cleavage signals involved in genome packaging [65]. These 2 DNA elements together could be sufficient for amplification *in situ* and for transmission to other cells if complemented by a 'helper' KSHV. Defective or deleted viral genomes in primary tumors have been reported previously for gamma-herpesviruses EBV [66,67] and KSHV [49], although their role in a natural infection is unclear. In KSHV, cells harboring defective KSHV genomes that had 82 kb deleted from the 5'end (including K5 and K6) had more malignant phenotypes than their parental BCBL-1 cell line but were lytic replication incompetent [49]. Methods such as RFLP or Southern Blot could be used to determine whether the elevated K5-K6 genome fragments detected here exist as defective viral genomes.

Two persons had identical KSHV genome rearrangements found in multiple tumors. Taken together with our earlier finding of such cases in 2 more individuals [30], this constitutes strong evidence that tumor-associated, rearranged KSHV genomes can spread by metastases, helper virus activity or residual infectivity of the rearranged genome. Identical KSHV genome rearrangements in distinct tumors implies that the viruses were clonally related, because independently acquiring the same breakpoint sites is highly improbable.

## Association of breakpoints with GC-rich DNA features

Certain patterns or compositions of DNA sequences that are conducive to non-B DNA structure formation can be hotspots of genomic fragility and rearrangement [43]. Here, the

positions of KSHV genome rearrangement breakpoints were associated with predicted non-B-DNA structures. There was clustering of KSHV rearrangement breakpoints in K4.1-K4.2, IR1, the minor repeat region, and in ORF18-ORF19. In the last cluster, 4 unique breakpoints were within 1.2 kb, with 2 breakpoints only 3 bp apart. IR1 and the minor repeat region are composed of tandem repeats with poly-G or C runs. Such repeats are prone to forming G-quadruplexes (G4), stable structures formed by 4 consecutive guanines folding into a tetrad and stacking with other guanine tetrads on the same strand [47,68,69]. ORF19 has a high-probability Z-DNA-forming sequence, precisely at the 2 breakpoints that were 3 bases apart. Z-DNA refers to a left-handed helix in a zigzag pattern formed by alternating purines and pyrimidines [69]. Z-DNA and G4 can form during replication, transcription and other events that generate negative supercoiling and expose single-stranded DNA [69]. When left unresolved, they can interfere with the progression of replication forks, leading to double-stranded DNA breaks [69].

G4 were the most common among the five non-B DNA structures predicted within 500 bp of the 31 KSHV rearrangement breakpoints identified here and in other studies [30,42,49]. G4 had significantly higher summed scores and shorter distances to the observed breakpoints, compared to 1,000 simulations of 31 random points on the GK18 genome. This result is consistent with G4 being associated with recombination breakpoints in other herpesviruses [45,70]. GC-rich DNA motifs that facilitate double-stranded breaks for recombination-dependent repair are enriched in gamma-herpesvirus genomes beyond what is expected from their nucleotide composition, and compared to alpha- and beta-herpesviruses [44].

## Novel Intra-host and Inter-host Variations

K8.1 is an envelope glycoprotein necessary for infecting primary B-cells but not primary endothelial cells [71,72]. Here, nine different K8.1 inactivating mutations were found in 8 of 29 individuals–including nonsense mutations, promoter region deletions and rearrangement breakpoints. The nonsense mutations observed here and in other studies truncate the K8.1 coding sequence, removing its C-terminal transmembrane anchor domain [73].

K8.1 was the only KSHV gene with inactivating mutations found in more than one individual. To date, inactivating mutations in K8.1 have been observed only in KSHV genomes derived from KS tumor biopsies (GK18, ZM124 and Miyako1), and not in matching oral swab samples [20,30]. K8.1 mutations were also often tumor site-specific since one person had 2 different K8.1 mutations and other tumors from the same individuals often had intact K8.1. Hence, selection against K8.1 expression may be local.

K8.1 is highly immunogenic [73–76]. However, unlike the lytic membrane protein K1 that has hypervariable extra-cellular domains, full-length K8.1 sequences are highly conserved *across* individuals. Additionally, the C-terminal region of K8.1 that is often truncated are not among known antigenic hotspots [77,78]. Given that the K8.1 mutations observed were inactivating, tumor-associated, and unique to every individual, they must arise *de novo*. Characterization of the immune-reactivity of individuals harboring KSHV with an inactivated K8.1 gene will be needed to substantiate the hypothesis of immune driven selection.

KSHV microRNA-K10 is weakly transforming [79], and a single nucleotide change is known to alter its tumorigenicity [80]. Its 23-nt U<u>AGUGUUGUCCCCC</u>**CG**AGUGGCC sequence is tightly conserved in the vast majority of KSHV genomes evaluated to date [81]. The C15T and G16A mutations observed in this study (both bolded above) were found outside the miR-K10 seed sequence (underlined). These mutations were also tumor-specific and are predicted to result in a slightly more stable stem loop [30]. The miR-K10 mutations were also almost always the only intra-host synonymous mutation detected in the ~131 kb of the KSHV genomes sequenced.

The K4.2 and K11 genes were polymorphic across individuals, although no intra-host differences were found. Among the 96 KSHV genomes sequenced to date, K4.2 had 3 length classes: the prototypic 549-bp full-length form, a 378 or 369-bp truncated form, and a 237-bp or shorter truncated form. A majority of KSHV genomes from Africa, now comprising most of the KSHV genomes sequenced to date, carry the truncated K4.2 forms. The K11.2 polymorphism corresponds to a 174-bp duplication in its central domain, and present in 76 of 94 of the known whole KSHV genomes. K4.2 and K11.2 polymorphisms do not coincide with the phylogenetic groupings of KSHV genomes, although all representatives of the genome type P2 have the 237-bp class of K4.2.

The full-length K4.2 gene expresses an immediate-early protein that inhibits endoplasmic reticulum chaperone protein pERP1, enhancing KSHV glycoprotein maturation among other effects [82]. All K4.2 truncations delete its putative transmembrane domain, which abrogates localization of K4.2 to the ER [82]. It has been reported that completely deleting K4.2 diminishes the expression of KSHV glycoproteins K8.1 and gB, dampening infectious virus titer by 4-fold [82]. This effect on K8.1 production is noteworthy in the context of our findings, because K4.2 truncation and K8.1 inactivation were inversely correlated, i.e., they tended to not co-occur in this cohort (p = 0.013). Suggestively, truncated K4.2 may have the same effect as a K8.1 inactivation in depressing K8.1 production. This can be tested by immunohistochemistry for surface expression of intact K8.1 in KS tumors with and without the K4.2 truncations. Furthermore, according to this hypothesis, K8.1 levels would be diminished or absent in individuals with truncated K4.2.

K11.2 encodes a viral homolog of cellular interferon regulatory factor 2 (vIRF-2), which modulates the antiviral signaling of interferon [83,84] and regulates KSHV lytic replication by suppressing KSHV early lytic gene expression [85]. While the C- and N-terminal domains of vIRF-2 has been characterized [85,86], the function of the central 174-bp repeat has not been elucidated. The repeated sequence may contain internal translation initiation sites that produce the shorter vIRF-2 isoforms that have been observed [85].

## Clinical phenotypes of mutations

K5-K6 region overrepresentation and K8.1 inactivation were more likely in nodular rather than macular tumors (p<0.001 and p = 0.010 respectively), while it was the opposite for K4.2 truncations (p = 0.028). The latter argues against the hypothesis above, that the effect of K4.2 truncation and K8.1 inactivation may have the same biological effect. Persons with more widespread lesions (>4 of 8 anatomic areas) were less likely to have K5-K6 region overrepresentation (p = 0.01) and K8.1 inactivation (p = 0.024). miR-K10 mutations were rarer in tumors <1 cm in diameter (p = 0.010) and with a trend to lower survival rate (p = 0.053). However, since most study participants were HIV-positive, the strong effect that active HIV replication has on KS development would likely override any weak association that could exist between KSHV genome alterations and disease characteristics. A similar study on adults with HIV-negative endemic and classic KS may strengthen this association.

Study participants entered the study at varying stages of their illness, so there is heterogeneity in the tumor stages sampled at baseline as well as in the clinical course following admission. While we conducted a large sampling, we examined only a minority of tumors found in many individuals, implying that we are providing only a minimum estimate of tumor-associated intra-host mutation frequencies. Nonetheless, even the most common tumor-associated viral mutations observed, K5-K6 region overrepresentation and K8.1 inactivations were still only found in a minority of KS tumors sampled. As such it follows that the tumor associated KSHV mutations were not required for tumorigenesis, and instead may have been a common

outcome of KSHV replication in tumor cells. However, their frequency and clinical associations suggest that these mutations could be influential during KS tumor development. Our findings raise some hypotheses on the potential of these genes to contribute to KS pathology *in vivo*, a study of which can inform clinical treatments and management of the disease.

## Supporting information

**S1 Fig. PCR and sequencing confirmation of rearrangement connecting IR1 and K6 sequences in U048-D. A.** Primers (blue arrows) used to PCR across breakpoint junctions in the tumor DNA extract is shown below a KSHV genome section (GK18 numbering) from K4.2 through IR1 and K6. ORFs are in yellow, repetitive sequences in orange. **B.** PCR products from indicated primers separated on a 0.8% agarose gel. Visible bands were extracted and sequenced. DNA from the KSHV-infected BCBL1 cell line was used as control. **C.** Results from sequencing the numbered bands in B. Sequencing of bands 2a and 2c using IR1 primers from both ends unexpectedly show reads mapping to K6 sequences. Sequencing for band 2b failed repeatedly, perhaps due to GC content. Bands 3 and 4b, which are not present in BCBL1, show that K6 sequences are connected directly to IR1 sequences at ~1 kb or less, suggesting an insertion. Band 4a shows the normal expected size between IRcomplex-F and K6-R, as seen in BCBL1.
(TIF)

**S2 Fig. Inversion between ORF9 and ORF10 in U048-D.** An 82 bp inversion was detected from coordinates 14,481 to 14,562. Below are Sanger sequence reads aligned to reference. Black marks represent mismatches to the reference, and the height of the chromatogram indicates the quality of the read. The gap next to the inversion represents an 8 bp deletion.
(TIF)

**S3 Fig. The probabilities and relative scores of 5 non-B-DNA structures.** Cruciform, local melt region, Z-DNA, cruciform, G-quadruplex and triplex DNA at individual clusters of breakpoints are shown. No cruciform was found. G-quadruplex scores are normalized to 300, near the human genome maximum (https://pqsfinder.fi.muni.cz/genomes), and triplex scores are normalized to a maximum of 50.
(TIF)

**S1 Table. Primer sequences used in this study.**
(XLSX)

**S2 Table. Breakpoints identified in KSHV genomes.**
(XLSX)

**S3 Table. PCR screening and sequencing results in oral swabs.**
(XLSX)

**S4 Table. K4.2 and K11.2 polymorphisms and KSHV genome types.**
(XLSX)

**S5 Table. Clinical characteristics of cohort participants.**
(XLSX)

**S6 Table. KS tumor characteristics and morphotypes.**
(XLSX)

**S7 Table. Odds ratio of participant and tumor characteristics with KSHV mutations from logistic regression analysis of KS tumors from 29 participants, univariable analysis.**
(XLSX)

**S8 Table. Correlations of observed KSHV mutations.**
(XLSX)

**S9 Table. KSHV genetic polymorphisms on survival rates.**
(XLSX)

**S1 Alignments. SA1. Alignment of K4.2 sequences found in 96 published KSHV genomes. SA2. Alignment of K11.2 sequences found in 96 published KSHV genomes.**
(ZIP)

**S1 Graphs. 30 graphs of probabilities and scores of predicted non-B-DNA structures 500 bp before and after individual breakpoints examined in the manuscript.** The coordinate numbers in the figures of some breakpoints differ slightly from those in Column D of S2 Table because working draft genomes were used for non-B-DNA analyses, while S2 Table lists coordinates in the finished genomes uploaded to Genbank. The coordinate differences come from refinements in the genome termini and repeat regions, outside the analyzed sequences.
(ZIP)

## Author Contributions

**Conceptualization:** Jan Clement Santiago, Warren Phipps.

**Data curation:** Jan Clement Santiago, Scott V. Adams, Fred Okuku.

**Formal analysis:** Jan Clement Santiago, Scott V. Adams.

**Funding acquisition:** Warren Phipps, James I. Mullins.

**Investigation:** Fred Okuku, James I. Mullins.

**Methodology:** Jan Clement Santiago, Scott V. Adams, Andrea Towlerton, Fred Okuku.

**Resources:** Andrea Towlerton, Warren Phipps, James I. Mullins.

**Supervision:** James I. Mullins.

**Validation:** Jan Clement Santiago.

**Visualization:** Jan Clement Santiago, Scott V. Adams.

**Writing – original draft:** Jan Clement Santiago.

**Writing – review & editing:** Jan Clement Santiago, Scott V. Adams, Andrea Towlerton, Warren Phipps, James I. Mullins.

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
