## [Decision Letter · Decision Letter 0]

25 Jun 2022

Dear Mullins,

Thank you very much for submitting your manuscript "Genomic changes in Kaposi Sarcoma-associated Herpesvirus and their clinical correlates" for consideration at PLOS Pathogens. As with all papers reviewed by the journal, your manuscript was reviewed by members of the editorial board and by several independent reviewers. In light of the reviews (below this email), we would like to invite the resubmission of a significantly-revised version that takes into account the reviewers' comments.

We cannot make any decision about publication until we have seen the revised manuscript and your response to the reviewers' comments. Your revised manuscript is also likely to be sent to reviewers for further evaluation.

Sincerely,

Denise Whitby

Guest Editor

PLOS Pathogens

Klaus Früh

Section Editor

PLOS Pathogens

Kasturi Haldar

Editor-in-Chief

PLOS Pathogens

orcid.org/0000-0001-5065-158X

Michael Malim

Editor-in-Chief

PLOS Pathogens

orcid.org/0000-0002-7699-2064

Reviewer's Responses to Questions

**Part I - Summary**

Reviewer #1: This is an interesting paper by Santiago and others to study the KSHV genome variations using NGS of the whole genome of the Kaposi’s sarcoma herpesvirus (KSHV) of tumor DNA obtained from a cohort of patients from Uganda. Sixty-five tumors from 29 individuals were analyzed in this study. The authors found that there were frequent over coverage by ddPCR and sequence reads that the region covering K5 and K6 have higher coverage than the rest of the KSHV genome from the same tumors, suggesting multiple copies of the genes within the genome, supporting their previous findings. In addition, there were alternations, rearrangements and mutations in the K8.1 and the miR-K10 microRNA coding sequences.

This is a very detailed and extensive study as a follow up of a study with a smaller number of tumors by the same group and several studies from other groups. The results presented here supported most of their earlier findings. This study is interesting and potentially important because KSHV genome variations and de novo mutation may contribute the biology of the virus and disease pathogenesis. However, there are a number of concerns with the study and its analysis. In addition, there are some discrepancies throughout.

1) A major concern with the study is over representations, differential copy numbers and variations in a number of KSHV genes from the same tumor, as indicated by both ddPCR and genome sequencing. As shown in Table 2, there are substantial differences in the copy numbers between K2, ORF16, ORF50 or ORF73. For sample U020-E, there was only 1 copy of ORF50, but 172 copies of ORF73. It will be difficult to imagine that they could exist within the same genome. There is a need to demonstrate that such duplication occurs in the same genome in order to support the biological significance of gene duplication in the biology of the virus and tumorigenesis.

2) The mutations in the miR-K10, and deletions and rearrangements of K8.1 genes are consistent and support other previous studies, it is possible that they could be an outcome of tumor cell and KSHV genome replication, rather than they playing a causative role in KS development.

3) The informatics workflow of the study was very informative, yet they could be more detailed and supported, based on the numerous tools that were used, and the fact that the KSHV genomes were de novo assembled.

4) Their claims were not always supported by their figures. For instance, where they claim "G4 scores also peaked at the breakpoint position but not in the control data", however, that does not seem to be the case in the figure for regions in Fig 4D (roughly) around 225-250 nt.

5) What is the rationale behind their associations between non-B DNA structures (through their predicted sequence features) and rearrangement breakpoints? It could be better to calculate an overall predicted Z-DNA, such that, instead of permuting the breakpoint sequence interval, cruciform, melt motifs, etc., they could pick random regions of the same length that are not within rearrangement breakpoints, and then compare. It would then be useful to see if these features are really overrepresented in the rearrangement breakpoint sequences relative to non-associated real biological sequences. What about cruciforms in these regions and in the overall KSHV genome?

6) In addition, more careful editing and attention to details will be needed. The errors made it difficult for this reviewer and the readers to follow. For example:

(i) Table numbers seem to be incorrect throughout the result section. Table 2 should be 1, Table 3 should be 2 etc. There should be only 3 tables instead of Table 4 as referred to in the result section.

(ii) There were 28 supplementary graphs as opposed to 31 mentioned in the manuscript.

(iii) Fig. S2 does not match its caption.

(iv) There were acronyms not described in the order of their mention throughout the manuscript;

(v) There were typos, even in the abstract.

Reviewer #2: In this very interesting manuscript, Santiago and colleagues present the so far largest set of full-length genomic KSHV sequences obtained from KS (Kaposi Sarcoma) tumors. Their findings highlight that rearrangements and duplications of parts of the KSHV genome can occur in tumors and differ between multiple tumors observed in the same patient. They also catalogue missense mutations that occur in two KSHV genes, K4.2 and K8.1, and provide evidence that these arise during the development of individual KS lesions. Furthermore they attempt to correlate the presence of certain mutations with the development of KS disease.

The strength of this manuscript lies in the large scale and systematic analysis of KSHV genomic rearrangements and mutations. Their findings extend previous findings made in cell lines to tumors harvested from patients and thereby illustrate the biological relevance of some of the KSHV genome alterations reported in their manuscript. A weakness is that they attempt multiple statistical comparisons of individual KSHV genomic rearrangements or mutations in individual genes with tumor size, disease outcome etc., without correcting for multiple statistical comparisons.

Reviewer #3: The authors present a comprehensive analysis of genomic features of Kaposi Sarcoma-associated Herpesvirus (KSHV) genomes that have been isolated from tumors. This analysis includes data from a prior publication of an initial 12 viral genomes from the same dataset. Those data are now supplemented by an additional 20 viral genomes. The introduction is superbly written, with an excellent coverage of the prior relevant literature. The analyses are thorough and clear, and the patterns observed will no doubt move the field forward. Minor areas of clarification noted below would improve this already-excellent manuscript.

**Part II – Major Issues: Key Experiments Required for Acceptance**

Reviewer #1: Please see above.

Reviewer #2: 1. Figure 2: the authors should include a figure or table that lists the exact breakpoints of the amplified genomic regions, shown schematically in figure 3, that they found by Sanger sequencing of PCR products. Figure 3 does not allow to deduce these exact breakpoints.

2. The multiple statistical comparisons of KSHV genomic features (amplifications, duplications, point mutations, protein truncations) with disease progression, tumor stage, etc. were done without correcting for multiple comparisons. While probably strictly not appropriate, such an approach can serve to tentatively identify genomic features that might influence disease characteristics and that could be followed up in later studies. The authors must therefore stress the tentative and explorative nature of these analyses in the abstract. They should also comment (in the discussion) on the fact that most of their KS tumors were from HIV-positive patients and that the strong effect that active HIV replication has on KS development would likely override any weak association that could exist between alterations in the KSHV genome and disease characteristics. It would be of interest to conduct a similar study in patients with HIV-negative endemic KS (in Africa) or classic KS (in Southern Europe).

3. Figure S5: the two groups analyzed here (WT miRK10, mutated miRK10) are of very different size, which could complicate the Kaplan-Meier analysis shown here. The Kaplan Meier curve yielded a p value of 0.038 (figure S5), whereas the Cox regression analysis presented in table S9 came to a p value of 0.053, which is also mentioned and discussed in the text. The authors should decide which analysis they want to include (probably the Cox regression) and omit the other analysis.

Reviewer #3: - Figure S3 provides a valuable overview of the present study set of viral genomes and how these relate to other published sequences. The authors should consider moving this data into the main manuscript and integrating the data from Figure 6C onto this SplitsTree. Regardless of where it appears, please highlight on the SplitsTree the 65 viral genomes analyzed in this study. Otherwise the similarly named “UG..” strains are hard to distinguish from the present study’s “U..” strains.

- Following on the above, Figure S3 clearly highlights the major clades within the KSHV tree. As illustrated in Figure 6C & Table S4, the clades do not match perfectly with the three versions of K4.2 variants or K11.2 repeats. Do any of the other variants found in multiple individuals in the present study (e.g. K8.1 or K5-K6 expansions) show any segregation on the SplitsTree? While these variants can clearly arise independently and de novo (e.g. Figure 3), there may still be an interaction with the genetic background of certain clade(s), such that these mutations are more likely to occur in one type of strain background(s) than another.

- There are several opportunities to better integrate data across multiple figures or analyses. For example, Figure 1 has the only full-genome diagram in the manuscript. This makes it a valuable resource to refer to throughout the paper, but at present it only highlights genes of interest to Figure 1. Please consider highlighting additional genes or regions of interest from later figures on here as well (in a different color or row if need be)? E.g. K8.1, mir-K10, K4.2, K11.2, K5, K6, etc.

- Table 1. Figure 1 and section starting on line 218 – can the authors please clarify the degree to which ddPCR and genome-sequencing did or did not agree, with regard to over-coverage of the region around IR1? The table of KSHV genome characteristics shows the ddPCR estimation of viral genomes with over-coverage around IR1 (column 7). For most of the “bolded” ratios, Figure 1B shows a good match in over-coverage by genome-sequencing to these ddPCR estimations. However the matchups do not work for viral genomes U215-D, U156-D, or U108-B – can the authors comment on why this might be, or which technique they feel is more accurate? Other samples from their prior study (ref [30]) are bolded in Table 1 but not shown in Figure 1B – how did those genomic-coverage plots match up with ddPCR estimations of duplications? These data may suggest which method(s) should be used in future studies, so this comparison is relevant to explore.

- Lines 493-498 – Can the authors speculate or point to any data on whether or not the K5-K6 over-coverage areas – and their rearrangements – might indicate that the tumors contain sub-genomic fragments containing just these regions with IR? If so, is this an in vivo correlate of defective interfering (DI) genomes or particles (DIPs), aka defective viral genomes (DVGs)? While it may well be beyond the scope of this paper to do so, is there sufficient viral DNA that methods such as RFLPs or Southern blot(s) could be used to help figure this out?

**Part III – Minor Issues: Editorial and Data Presentation Modifications**

Reviewer #1: (i) Table numbers seem to be incorrect throughout the result section. Table 2 should be 1, Table 3 should be 2 etc. There should be only 3 tables instead of Table 4 as referred to in the result section.

(ii) There were 28 supplementary graphs as opposed to 31 mentioned in the manuscript.

(iii) Fig. S2 does not match its caption.

(iv) There were acronyms not described in the order of their mention throughout the manuscript;

(v) There were typos, even in the abstract.

Reviewer #2: 1. Lines 246, 257, 261, 298, 420: 'Table 2' should probably be 'Table 1'.

2. Figure 5A: the meaning of the color code used for the vertical lines in the boxes denoting individual ORFs should be explained in the legend.

3. Figure 5B: for sample UO20-E, in the line 'Translation', there should be an 'H' to indicate the coding change in the protein sequence (as for samples U004-D, 156-D).

4. Line 461: the Kaplan Meier curve for the impact of the mirK10 mutation on disease progression is shown in figure S5, not S4;

5. lines 613ff: The legends for figures S2-S6 are either not corresponding to the individual supplementary figures of are missing.

Reviewer #3: Minor fixes:

- Tables & Supplemental Figures are mis-numbered and need to be updated. There are two Table 1’s, and subsequent numbering of Tables (though not their in-text references) are off. For the Supplemental Figures, Fig S1B should be Fig S2 and so on.

- For Figure 2 and Figure S1, is it possible to provide a diagram of the inferred rearrangement, to complement the canonical genome image and the primers? This would help to visualize how the bands match the “Sequence Interpretations” in the embedded figure-tables.

- Line 121 – how many cycles of PCR enrichment at this step?

- Lines 149-152 – Is there any reason why read data is aligned to both the human and KSHV genomes for LUMPY analysis (breakpoint detection)? If so, it would be good to add a comment on this aspect for other researchers to consider.

- Line 206 – any reason for the strong male bias in the study population? Any expected impact of this on the correlations observed (e.g. with disease manifestation(s))?

- Lines 205-209 – this part is quite short. It would be helpful to summarize here how the 67 tumor biopsies relate to the 29 individuals, e.g. how many people had multiple biopsies, and how many were single-sample only? How well does this availability map to the viral genomes in Table 2?

- Table 1 – for the first numeric column, it’s not easy to distinguish which are “N” and which are “median”. Can these be indicated better?

- Figure 1B – please add dash to sample names (e.g. U219-D instead of U219D) to sync with the text and table

- Figure 3 – please fix yellow and white text labels on the genome diagram; they are hard to read at present.

- Table 1, Table S4, Figure 6 – please synchronize the naming of the K4.2 length variants to be the same across text, figure, and tables. At present, Table S4 refers to Truncation A or B, while Table 1 gives only the length, and the Figure offers both options.

- Line 368 – should “intra”-host here be “inter” instead?

- Figure 5 – using gray bars for both the DNA (genome) and protein (translation) lines here makes the figure hard to read. Recommend changing one or the other, and then matching across other figures (e.g. in Figure 6, translations are gray bars, and in Figure 3, high-coverage areas are also gray bars).

- Table 3 (supposed to be 4) on “Coding sequence mutations” – the footnote here is critical to the interpretation of the table and should be integrated into the table’s title.

- Figure 6C – was this tree made using full-length genomes, or just the K4.2 and K11 regions?

- Figure 7 – please add explanation on arrows to the figure legend

PLOS authors have the option to publish the peer review history of their article (what does this mean?). If published, this will include your full peer review and any attached files.

Reviewer #1: No

Reviewer #2: **Yes: **Thomas F. Schulz

Reviewer #3: No
---

## [Decision Letter · Decision Letter 1]

10 Oct 2022

Dear Mullins,

Thank you very much for submitting your manuscript "Genomic changes in Kaposi Sarcoma-associated Herpesvirus and their clinical correlates" for consideration at PLOS Pathogens. As with all papers reviewed by the journal, your manuscript was reviewed by members of the editorial board and by several independent reviewers. The reviewers appreciated the attention to an important topic. Based on the reviews, we are likely to accept this manuscript for publication, providing that you modify the manuscript according to the review recommendations.

Sincerely,

Denise Whitby

Guest Editor

PLOS Pathogens

Klaus Früh

Section Editor

PLOS Pathogens

Kasturi Haldar

Editor-in-Chief

PLOS Pathogens

orcid.org/0000-0001-5065-158X

Michael Malim

Editor-in-Chief

PLOS Pathogens

orcid.org/0000-0002-7699-2064

Reviewer Comments (if any, and for reference):

Reviewer's Responses to Questions

**Part I - Summary**

Reviewer #1: The reviewers have modified the manuscript to try to address most of the concerns. They have also addressed the specific informatics comments on their control randomized sequences for comparison to the tumor genome sequences. However, it is still not fully transparent on their data analysis. Majority of their highlighted sections up until Results were unchanged, just emphasized those for the reviewers. They seem to have fixed the captions and table legends and the number of figures etc.

One very minor comment, I think they need to cite Geneious Prime, and mention the version they are using with a URL, as instructed here (https://help.geneious.com/hc/en-us/articles/360044627352-How-do-I-cite-Geneious-or-Geneious-Prime-in-a-paper-)

Reviewer #3: As noted in my initial review, the authors’ manuscript on KSHV genomic analyses directly from tumor tissue is well written, thorough and clear. These data advance the field and raise new questions that future studies will seek to explore.

**Part II – Major Issues: Key Experiments Required for Acceptance**

Reviewer #1: (No Response)

Reviewer #3: None noted.

**Part III – Minor Issues: Editorial and Data Presentation Modifications**

Reviewer #1: (No Response)

Reviewer #3: The authors have answered the questions that I raised, as well as those of the other two reviewers. The minor clarifications we suggested during peer review have all been addressed. This has produced a valuable and clear product for future readers.

PLOS authors have the option to publish the peer review history of their article (what does this mean?). If published, this will include your full peer review and any attached files.

Reviewer #1: No

Reviewer #3: No

Figure Files:

Data Requirements:

Reproducibility:

References:

---

## [Editor Report · Decision Letter 2]

7 Nov 2022

Dear Mullins,

We are pleased to inform you that your manuscript 'Genomic changes in Kaposi Sarcoma-associated Herpesvirus and their clinical correlates' has been provisionally accepted for publication in PLOS Pathogens.

Best regards,

Denise Whitby

Guest Editor

PLOS Pathogens

Klaus Früh

Section Editor

PLOS Pathogens

Kasturi Haldar

Editor-in-Chief

PLOS Pathogens

orcid.org/0000-0001-5065-158X

Michael Malim

Editor-in-Chief

PLOS Pathogens

orcid.org/0000-0002-7699-2064
---

## [Editor Report · Acceptance letter]

20 Nov 2022

Dear Mullins,

We are delighted to inform you that your manuscript, "Genomic changes in Kaposi Sarcoma-associated Herpesvirus and their clinical correlates," has been formally accepted for publication in PLOS Pathogens.

Best regards,

Kasturi Haldar

Editor-in-Chief

PLOS Pathogens

orcid.org/0000-0001-5065-158X

Michael Malim

Editor-in-Chief

PLOS Pathogens

orcid.org/0000-0002-7699-2064